



**Variations in diurnal and seasonal net ecosystem carbon dioxide**
**exchange in a semiarid sandy grassland ecosystem in China's Horqin**
**Sandy Land**
Yayi Niu[a,b,c,e], Yuqiang Li[a,b,c,*], Hanbo Yun[a,d,e], Xuyang Wang[a,b,c], Xiangwen Gong[a,b], Yulong
Duan[a,b,c], Jing Liu[a]
[a] Northwest Institute of Eco-Environment and Resources, Chinese Academy of Sciences, Lanzhou
730000, China
[b] University of Chinese Academy of Sciences, Beijing 100049, China
[c] Naiman Desertification Research Station, Northwest Institute of Eco-Environment and Resources,
Chinese Academy of Sciences, Tongliao 028300, China
[d] State Key Laboratory of Frozen Soil Engineering, Northwest Institute of Eco-Environment and
Resources, Chinese Academy of Sciences, Lanzhou, Gansu 730000, China
[e] Center for Permafrost (CENPERM), Department of Geosciences and Natural Resource
Management, University of Copenhagen, DK-1350 Copenhagen, Denmark
*Correspondence to*: Yuqiang Li (liyq@lzb.ac.cn)



**Abstract**

Grasslands are major terrestrial ecosystems in arid and semiarid regions, and play important roles in the regional carbon dioxide ($CO_2$) balance and cycles. Sandy grasslands are sensitive to climate change, yet the magnitudes, patterns, and environmental controls of their $CO_2$ flows are poorly understood. Here, we report the results from continuous year-round $CO_2$ observations in 5 years from a sandy grassland in China's Horqin Sandy Land. The sandy grassland was a net $CO_2$ source at an annual scale, with a mean annual net ecosystem $CO_2$ exchange (NEE) of $48.88 \pm 8.10$ g C m$^{-2}$ yr$^{-1}$ in the years for which a complete dataset was available (2015, 2016, and 2018); total annual precipitation was the most important factor for NEE. At a seasonal scale, the sandy grassland showed net $CO_2$ absorption during the summer, but net $CO_2$ release in the other seasons. The main environmental factors that affected NEE were temperature and soil water content (SWC) in the spring, soil heat flux and solar radiation in the summer, soil heat flux and temperature in autumn, and SWC and temperature in winter. At the diurnal scale, net solar radiation was the most important factor for NEE in all seasons. The sandy grassland may have been a net annual $CO_2$ source at an annual scale because the study site is recovering from degradation, thus vegetation productivity is still low. Therefore, the ecosystem has not yet transitioned to a $CO_2$ sink and long-term observations will be necessary to reveal the true source or sink intensity and its response to environmental and biological factors.

**Keywords:** Net ecosystem $CO_2$ exchange (NEE); Gross primary productivity (GPP); ecosystem respiration ($R_{ec}$); Eddy covariance; Horqin Sandy Land

**1 Introduction**

Arid and semiarid ecosystems cover 30 to 40 % of the global terrestrial surface (Poulter et al., 2014). The extent and distribution of these areas are increasing in response to factors such as climate change, changes in wildfire frequency and intensity, and changes in land use (Asner et al., 2003; Hastings et al, 2010). These ecosystems are important because they account for 30 to 35 % of terrestrial net primary productivity (Gao et al., 2012; Liu et al., 2016a) and approximately 15 % of the global soil organic



carbon pool (Lal, 2004; Liu et al., 2016a). Thus, the high potential carbon sequestration
in arid and semiarid areas may have a greater impact on the future global carbon cycle
than sequestration in tropical rainforest areas (Emmerich, 2003; Nosetto et al., 2006;
Poulter et al., 2014), and arid and semiarid ecosystems will have significant effects on
the global carbon cycle and carbon balance (Lal, 2004). However, arid and semiarid
ecosystems have received much less attention than wetter ecosystems, and the lack of
high-frequency continuous observations has led to a weak understanding of their role
in global terrestrial net ecosystem $CO_2$ exchange (NEE) (Baldocchi et al., 2001;
Hastings et al., 2010).
Desertification occurs in more than two-thirds of the area of arid and semiarid
ecosystems (Lal, 2001). This may cause a serious imbalance in the structure and
function of these ecosystems (Huenneke et al., 2002; Vest et al., 2011), especially in
terms of whether the ecosystem functions as a carbon source or sink (Shachak et al.,
1998; Gang et al., 2011). Grazing exclusion is a common method used to combat
desertification in the world's arid and semiarid areas (Mureithi et al., 2010; Sousa et al.,
2012). For example, Sun et al. (2015) suggested that proper exclosures promoted the
recovery of degraded sandy grassland and more sustainably use sandy grassland
resources.
The Horqin Sandy Land is the largest sandy land in China, and nearly 80% of the
area has been desertified (Li et al., 2019). The sandy land includes multiple overlapping
ecotones, including transition zones between areas with different population pressures,
between semi-humid and semiarid areas, and typical agro-pastoral ecotones. The
ecological environment is fragile and extremely sensitive to climate change and human
activities (Bagan et al., 2010; Zhao et al., 2015). The region's sandy grassland grows
on aeolian sandy soils or with sandy soils as the substrate, and is typical of the grassland
vegetation that develops in sandy land. This grassland ecosystem is widespread in the
Horqin Sandy Land (Munkhdalai et al., 2007; Zhao et al., 2007). Research showed that
the restoration of degraded sandy grassland can increase its productivity and carbon
sequestration, and that the ecosystem can begin to act as a carbon sink (Ruiz-Jaen and





Aide, 2005; Zhao et al., 2016). However, other studies showed that it was a carbon
source (Li et al., 2012; Niu et al., 2018). There have been relatively few long-term
studies of sandy grassland at the ecosystem level, so we do not yet fully understand the
characteristic of NEE and its components, gross primary productivity (GPP) and
ecosystem respiration ($R_{ec}$), at an ecosystem scale, particularly for sandy grassland
protected by grazing exclosures, and more data are needed, particularly for semiarid
sandy land (Barrett, 1968; Czobel et al., 2012).

Precipitation is one of the factors that most strongly affects NEE in arid and semiarid

areas. It affects NEE mainly through its effects on GPP and $R_{ec}$ (Lal, 2004; Dasci et al,
2010; Hastings et al, 2010; Liu et al., 2016b). Previous research showed that reduced
precipitation caused a continental-scale reduction in GPP, with concurrent decreases of
$R_{ec}$ (Ciais et al., 2005). In contrast, other studies found that GPP was more sensitive
than respiration to precipitation (Thomas et al., 2010; Shi et al., 2014). For instance,
Delgado-Balbuena et al. (2019) have shown that GPP was more than twice as sensitive
as $R_{ec}$ to precipitation in a semiarid grassland. The precipitation in the Horqin Sandy
Land is low, with high temporal and spatial variation, and other water resources such
as groundwater are small (Niu et al., 2015). Moreover, we found no reports on the
response of ecosystem-scale NEE and its components to precipitation, and the response
mechanisms are uncertain in the sandy grassland. Therefore, long-term data are
required to fully understand how changes in annual precipitation influence the annual
NEE, GPP, and $R_{ec}$ of sandy grassland.

In this paper, we present the results from continuous (14 September 2014 to 31

December 2018) *in situ* monitoring of $CO_2$ dynamics in the Horqin Sandy Land's sandy
grassland using the eddy covariance technique, and quantified the temporal variation of
NEE and the factors that control it. We had the following goals: (1) To quantify the
diurnal, seasonal, and annual variation in NEE, GPP, and $R_{ec}$. (2) To identify the
environmental factors that controlled NEE and its components at different temporal
scales, and the possible underlying mechanisms in the sandy grassland. (3) To
determine how annual precipitation affects the annual ecosystem NEE, GPP, and $R_{ec}$.



## 2 Materials and methods

### 2.1 Experimental site

Our study was conducted in a sandy grassland in the southern part of the Horqin Sandy Land, Inner Mongolia, China, at the Naiman Desertification Research Station of the Chinese Academy of Sciences (42 °55′ N, 120 °42′ E) (Fig. 1). The terrain is flat, and it evolved from reclamation of sandy grassland for agriculture to severe desertification, after which cultivation was abandoned and grazing exclosures were established to allow natural recovery of the vegetation, starting in 1985 (Zhao et al., 2007). Thus, the grassland had been recovering naturally for nearly 30 years when our study began. At an elevation of 377 m a.s.l., the study area has a continental semiarid monsoon temperate climate regime. The mean annual temperature is 6.8 ℃, with mean monthly temperatures ranging from -9.63 ℃ in January to 24.58 ℃ in July. Average annual precipitation is approximately 360 mm, with 70% of the precipitation occurring during the growing season, between June and August. Annual mean potential evaporation is approximately 1973 mm. The annual frost-free period is 130 to 150 days. The zonal soil is a sandy chestnut soil, but most of the soil has been degraded by a combination of climate change and anthropogenic activity (unsustainable grazing or agriculture) into an aeolian sandy soil under the action of wind erosion (Zhao et al., 2007), with coarse sand, fine sand, and clay-silt contents of 92.7, 3.3, and 4.0 % in the topsoil to a depth of 20 cm. The contents of soil organic carbon and total nitrogen were 1.27 and 0.21 g kg$^{-1}$, respectively. Vegetation cover in the study area ranged from 50 to 70 %. The dominant plant species were annual herbs, including *Artemisia scoparia, Setaria viridis, Salsola collina*, and *Corispermum hyssopifolium* (Niu et al., 2018).

### 2.2 Eddy covariance observations

An eddy covariance flux tower (2.0 m high) was installed at the center of the observation field (Fig. 1b, c). We have continuously monitored $CO_2$, water, and heat fluxes at the tower using the eddy covariance system since late 2014. The fetch from all directions was more than 200 m. Calculations with a footprint model indicated that the fetch was well within the desired flux footprint (Schmid, 1994; Xu and Baldocchi,



2004). The eddy covariance system consisted of an LI-7500 infrared gas analyzer (Li-
Cor Inc., Lincoln, NE, USA), with a precision of 0.01 μmol m$^{-2}$ s$^{-1}$ and an accuracy
within 1 % of the reading for measurements at 30-min mean intervals, and a CSAT-3
three-dimensional ultrasonic anemometer (Campbell Scientific, Inc., Logan, UT, USA),
with a precision of 0.1 ℃ and an accuracy of 1 % for the readings at 30-min mean
intervals. Raw 10-Hz data were recorded by a CR3000 datalogger (Campbell Scientific).
The operation, calibration, and maintenance of the eddy covariance system followed
the manufacturers' standard procedures. The LI-7500 was calibrated every 6 months
for $CO_2$, water vapor, and dew point values using calibration gases and dew point
generator measurements supported by the China Land–Atmosphere Coordinated
Observation System (Yun et al., 2018). We cleaned the mirror of the LI-7500 every 15
days to maintain the automatic gain control value below its threshold (55 to 65). All of
the instruments were powered by solar panels connected to a battery.

**2.3 Micrometeorological measurements**

Along with the flux measurements obtained by the eddy covariance equipment, we
measured standard meteorological and soil parameters continuously with an array of
sensors. A propeller anemometer was installed at the top of the meteorological tower to
measure the wind speed and direction. Net radiation ($R_n$, W m$^{-2}$) was measured by a
four-component radiometer (CNR-1, Kipp and Zonen, Delft, the Netherlands) installed
at 1 m above the ground. The air temperature ($T_{air}$, ℃) and relative humidity (%)
instruments (HMP45C, Vaisala Inc., Helsinki, Finland) were mounted at 2 m above the
ground to measure the $T_{air}$, relative humidity, and atmospheric pressure (kPa).
Precipitation (mm) measurements were obtained from a meteorological station 400 m
from the study site.
We installed five CS109 temperature probes (Campbell Scientific) and five CS616
moisture probes (Campbell Scientific) in the soil at depths of 10, 20, 30, 40, and 50 cm
to measure soil temperature ($T_{soil}$, ℃) and soil water content (SWC, %). Two self-
calibrating HFP01 soil heat flux (SHF, W m$^{-2}$) sensors (Hukseflux, Delft, the
Netherlands) were buried 5 and 10 cm below the ground to obtain the SHF data. All of



the environmental parameters were measured simultaneously with the eddy covariance
measurements, and all data were recorded as 30-min mean values with a CR3000
datalogger.
**2.4 Data quality and gap-filling method**
We used the EddyPro 6.2.0 software
(https://www.licor.com/env/products/eddy_covariance/software.html) to process the
10-Hz raw eddy covariance data. Processing included spike removal, lag correction,
secondary coordinate rotation, Webb–Pearman–Leuning correction, and sonic virtual
temperature conversion (Webb et al., 1980). We used the data processing method of Lee
et al. (2004) to process the 30-min mean raw flux measurements to ensure their quality.
Processed data were further corrected for weather effects and sensor uncertainty using
the following procedure: (1) We removed data gathered during precipitation events,
power failures, and sensor maintenance or malfunction. (2) We excluded unrealistic
$CO_2$ flux data (values outside the range of –2.0 to 2.0 mg $CO_2$ $m^{-2}$ $s^{-1}$). (3) We rejected
the data during stable atmospheric conditions at nighttime (friction velocity ($u_*$) < 0.1
m $s^{-1}$). Based on the $R_n$, NEE was classified as the daytime $NEE_{day}$ ($R_n \geq 1$ W $m^{-2}$) or
the night-time $NEE_{night}$ ($R_n < 1$ W $m^{-2}$). This screening resulted in the rejection of 20 to
30 % of the flux data, depending on the period.
We used several strategies to compensate for missing data. We used linear
interpolation to fill gaps that were shorter than 2 h. For longer gaps, we handled the gap
in the $NEE_{day}$ using the mean diurnal variation with a 7-day window (Falge et al., 2001),
and handled the gap in the $NEE_{night}$ using a temperature-dependent exponential model
(Lloyd and Taylor, 1994):
$NEE_{night} = R_0 \exp (b\ T_{10})$      Eq (1)
where $R_0$ is the base ecosystem respiration rate when the soil temperature is 0 °C, b
an empirically determined coefficient, and $T_{10}$ is the soil temperature at a depth of 10
cm. Daytime ecosystem respiration can be estimated by extrapolation from the
parameterization derived from Eq. (1). We did not attempt to fill in gaps longer than 7
days, and treated those gaps as missing data. Gross primary productivity (GPP) was



obtained as follows:
$GPP = NEE - R_{ec}$                                          Eq (2)
We evaluated the data quality based on the degree of energy closure (sensible heat +
latent heat – net radiation – soil heat flux). The energy closure values for the sandy
grassland from 2015 to 2018 were 86.5, 82.1, 57.7, and 85.2 %, respectively.
**2.5 Statistical analyses**
We performed correlation analysis (Pearson's r) and principal-components analysis
(PCA) using version 22 of the SPSS software (IBM, Armonk, NY, USA). Unless
otherwise noted, we defined statistical significance at $p < 0.05$. Pearson's r was applied
to confirm the strength of the relationships between parameters. PCA was used to
identify the main environmental factors that affected NEE and its components at
different temporal scales. Before performing PCA, we tested for collinearity (using a
variance inflation factor of $0 < VIF < 10$) using the Kaiser–Meyer–Olkin (KMO) test
and Bartlett's sphericity test. Collinearity was used to repartition the $T_{soil}$ and SWC data.
We considered KMO values $> 0.50$ and $p < 0.05$ for Bartlett's sphericity test to indicate
acceptable data (Hair et al., 2005). Our data were suitable for PCA, with the KMO value
ranging from 0.52 to 0.78 and $p < 0.001$ for all Bartlett's sphericity test results.
**3 Results**
**3.1 Meteorological conditions**
We recorded $T_{air}$ in the sandy grassland between September 2014 and December
2018 (Fig. S1). The mean annual $T_{air}$ was 7.38 °C, with minimum and maximum $T_{air}$ of
−17.82 and 27.15 °C, respectively. Average $R_n$ was 74.13 W m$^{-2}$, with average
minimum and maximum values for the 5 years minimum and maximum $R_n$ of −13.09
and 166.32 W m$^{-2}$, respectively (Fig. S2). The diurnal- scale SHF values at depths of 5
and 10 cm were 0.65 and −0.21 W m$^{-2}$, respectively (Fig. S3), with maximum values
of 34.43 and 26.88 W m$^{-2}$ (both on 28 April 2018), and the minimum values were –
36.26 and –23.77 W m$^{-2}$ (both on 4 October 2016).
The annual mean $T_{soil}$ values at depths of 10, 20, 30, 40, and 50 cm were 9.49, 9.66,
9.85, 10.22, and 10.65 °C (Fig. S4a). The average minimum diurnal values for the 5





years diurnal $T_{soil}$ at all depths were −18.08, −16.33, −14.17, −12.56, and −11.26 °C,
respectively, and the average maximum diurnal values for the 5 years diurnal $T_{soil}$ were
33.45, 32.31, 31.30, 30.49, and 29.76 °C, respectively. The annual mean SWC values
at depths of 10, 20, 30, 40, and 50 cm were 3.5, 3.6, 4.2, 4.8, and 5.7 %, respectively
(Fig. S4b). The average minimum diurnal values for the 5 years minimum diurnal SWC
at all depths were 1.5, 1.6, 1.6, 2.4, and 2.7 %, respectively, and the average maximum
diurnal values for the 5 years diurnal SWC were 10.8, 7.9, 11.0, 12.1, and 13.6 %,
respectively.
The site was windy, with an annual average wind speed of 3.19 m s$^{-1}$ from 2015 to
2018, and the principal direction of the strongest winds was from the northeast sector
(Fig. S5). In spring, the average wind speed was 2.25 m s$^{-1}$. In summer, the average
wind speed was about 2.51 m s$^{-1}$, predominantly driven by the north wind.
The annual precipitation totaled 212.3 mm in 2015, 276.8 mm in 2016, 312.8 mm in
2017, and 350.8 mm in 2018 (Fig. S4b). The mean annual precipitation was 288.2 mm,
which was less than the mean annual precipitation of 360.0 mm from 1960 to 2014.
During the spring (March, April, and May), precipitation was relatively abundant, with
mean precipitation of about 41.9 mm, which accounted for 12 to 20 % of the total annual
precipitation. The majority of the precipitation (about 65 %) occurred in the summer
(June, July, and August), with mean precipitation of about 197.1 mm. The autumn
(September, October, and November) precipitation was similar to that in spring, with a
mean precipitation of about 48.6 mm, which accounted for 14 to 23 % of the annual
total. During the winter (December, January, and February), the mean precipitation of
0.6 mm accounted for only 1 to 6 % of the annual total.
**3.2 Annual, seasonal, and diurnal variability of NEE, GPP and $R_{ec}$.**
Figure 2 suggests that the sandy grassland was a net $CO_2$ source, with an annual mean
NEE, GPP, and $R_{ec}$ of 48.88 ± 8.10, 351.78 ± 20.97, and 302.87 ± 28.96 g C m$^{-2}$ yr$^{-1}$ in
the years for which a complete dataset was available (2015, 2016, and 2018). (We
omitted 2017 from this calculation because of large gaps in the data, described below.)
NEE ranged from 34.99 C m$^{-2}$ yr$^{-1}$ in 2018 to 63.05 g C m$^{-2}$ yr$^{-1}$ in 2015, whereas GPP



ranged from 255.99 g C m$^{-2}$ yr$^{-1}$ in 2015 to 355.77 g C m$^{-2}$ yr$^{-1}$ in 2018 and R$_{ec}$ ranged
from 319.04 g C m$^{-2}$ yr$^{-1}$ in 2015 to 390.85 g C m$^{-2}$ yr$^{-1}$ in 2018. The eddy covariance
tower was set up in mid-August 2014, and the instrument was allowed to stabilize for
1 month before we began collecting data, from 15 September to 23 December 2014;
during that period, we measured a cumulative carbon release of 46.67 g C m$^{-2}$, with
cumulative GPP and R$_{ec}$ of 24.80 and 71.47 g C m$^{-2}$, respectively. From 15 February to
26 April 2017 and from 14 October to 6 November 2017, approximately 3 months of
data were missing due to instrument maintenance and calibration, and the cumulative
NEE, GPP, and R$_{ec}$ were 63.33, 273.67 and 337.00 g C m$^{-2}$ for the remaining 9 months
of the year. Note that the periods covered by the data are therefore not identical.
We also observed clear seasonal variations in the total seasonal NEE, GPP, and R$_{ec}$
(Fig. 3) and their diurnal cycles (Fig. 4). In spring, the sandy grassland was an
atmospheric $CO_2$ source in all years, with NEE, GPP, and R$_{ec}$ averaging 0.15 ± 0.05,
0.63 ± 0.03, and 0.78 ± 0.03 g C m$^{-2}$ d$^{-1}$, respectively (Fig. 3a). The diurnal NEE cycle
was characterized by a single absorption peak from around 07:30 to around 16:30 (Fig.
4a). Note that although all times in China are reported as the Beijing time, the study site
was not sufficiently far east of Beijing for this to affect the physiological meaning of
these times. The rest of the day was characterized by weak carbon absorption. The
average diurnal GPP was also characterized by a single peak from around 05:00 to
around 19:30, and the diurnal R$_{ec}$ was characterized by an approximately horizontal line
at about 0.78 g C m$^{-2}$ d$^{-1}$. The maximum and minimum average diurnal values for NEE
were 0.78 (20:00) and −0.81 (12:00) g C m$^{-2}$ d$^{-1}$, respectively, versus 1.57 (12:00) and
0.58 (4:30) g C m$^{-2}$ d$^{-1}$ for GPP and 1.01 (18:00) and 0.05 (17:30) g C m$^{-2}$ d$^{-1}$ for R$_{ec}$.
In summer, the sandy grassland was a $CO_2$ sink in all years, with NEE, GPP, and R$_{ec}$
averaging −0.58 ± 0.05, 2.44 ± 0.05, and 1.86 ± 0.03 g C m$^{-2}$ d$^{-1}$, respectively (Fig. 3b).
The diurnal cycles of NEE and GPP were also characterized by a single peak, between
05:00 and 19:30 (Fig. 4b), and the ecosystem $CO_2$ uptake reached its peak from around
10:30 to 12:00. The diurnal R$_{ec}$ pattern was similar to the spring, but at a higher level
(about 1.86 g C m$^{-2}$ d$^{-1}$). The maximum and minimum diurnal NEE averaged 2.67





(21:30) and −4.60 (11:30) g C m$^{-2}$ d$^{-1}$, respectively, versus 6.02 (11:30) and 0.09 (19:30)
g C m$^{-2}$ d$^{-1}$ for GPP and 2.77 (21:30) and 1.19 (8:00) g C m$^{-2}$ d$^{-1}$ for $R_{ec}$.
In autumn, the sandy grassland was a net source of atmospheric $CO_2$ in all years,
with NEE, GPP, and $R_{ec}$ averaging 0.50 ±0.02, 0.27 ±0.02, and 0.76 ±0.02 g C m$^{-2}$
d$^{-1}$, respectively (Fig. 3c). The diurnal dynamics of NEE, GPP, and $R_{ec}$ in autumn (Fig.
4c) were similar to those in spring (Fig. 4a), but the magnitudes of NEE and GPP in
autumn were lower than in the spring. The diurnal $R_{ec}$ was similar to the spring, at about
0.76 g C m$^{-2}$ d$^{-1}$. The maximum and minimum average diurnal NEE were 0.88 (19:00)
and 0.02 (11:30) g C m$^{-2}$ d$^{-1}$, respectively, versus 0.89 (17:30) and 0.63 (4:00) g C m$^{-2}$
d$^{-1}$ for GPP and 0.74 (12:00) and 0.01 (5:00) g C m$^{-2}$ d$^{-1}$ for $R_{ec}$.
In winter, the grassland ecosystem functioned as a net $CO_2$ source in all years, with
an average seasonal NEE of 0.58 ±0.01 g C m$^{-2}$ d$^{-1}$ (Fig. 3d). It should also be noted
that since the investigation started on 14 September 2014 and ended on 31 December
2018, the 2017 to 2018 winter was only about one-third of the usual length (i.e., it did
not include data from January and February 2019). The diurnal dynamics of the winter
NEE differed from the other seasons (Fig. 4d), with a minimum release value of 0.38 g
C m$^{-2}$ d$^{-1}$, and with two emission peaks: at 0.81 g C m$^{-2}$ d$^{-1}$ (08:00) and 0.89 g C m$^{-2}$
d$^{-1}$ (16:30).
**3.3 Response of NEE, GPP and $R_{ec}$ to changes in environmental factors**
We analyzed the effects of environmental factors on NEE and its components at
different temporal scales. The analysis methods for the diurnal scale (Pearson's *r* and
PCA) were the same as the methods used at a seasonal scale, so to avoid repetition, we
have only described the relationship between the seasonal-scale NEE and its
components and the associated environmental factors. At the diurnal scale, $R_n$ was the
main factor that affected NEE, GPP and $R_{ec}$ in all four seasons (data not shown). NEE
was significantly negatively correlated with $R_n$, whereas GPP and $R_{ec}$ were positively
correlated with $R_n$, indicating that the ecosystem's carbon sequestration capacity
increased with increasing $R_n$.
Seasonal-scale NEE, GPP, and $R_{ec}$ were significantly correlated with many





environmental factors (Table S1). We found extremely weak and non-significant
relationships between NEE, GPP, and $R_{ec}$ and two climate variables (relative humidity
and atmospheric pressure), so we excluded those variables from our subsequent
analysis. NEE was negatively correlated with most environmental factors in all seasons
except the autumn, when most correlations were positive, and GPP and $R_{ec}$ were
positively correlated with most environmental factors (Table S1).
In spring, $T_{soil}$ at all depths, and $T_{air}$, $R_n$, and SWC at all depths were negatively
correlated with NEE (Table S1). In the PCA, principal component 1 (PC1) explained
57 % of the NEE variation (Table 1), and was dominated by temperature ($T_{soil}$ at all
depths and $T_{air}$). PC2 explained about 25 % of the NEE variation, and was dominated
by SWC at depths of 0 to 10 cm. The first two PCs explained about 82 % of the NEE
variation. GPP was positively correlated with most environmental factors. PC1
explained 39 % of the GPP variation, and temperature and SWC at depths of 10 to 50
cm were the dominant factors (Table 2). PC2 explained about 33 % of GPP variation,
and was dominated by SHF at all depths. The first three PCs explained about 89 % of
the GPP variation. $R_{ec}$ was positively correlated with all environmental factors except
for wind speed. PC1 explained 42 % of the $R_{ec}$ variation (Table 3) and was dominated
by SWC at depth of 20 to 50 cm, $T_{soil}$ at all depths, $T_{air}$, and $R_n$. PC2 explained about
30% of the $R_{ec}$ variation, and was dominated by SHF at all depths. The first three PCs
explained about 89 % of the $R_{ec}$ variation.
In summer, PC1 explained 42 % of the NEE variation, and was dominated by SHF
at all depths and $R_n$ (Table 1). PC2 explained 29% of the NEE variation, and was
dominated by air and soil temperatures. The first three PCs explained about 88 % of the
NEE variation. For GPP, PC1 explained 36 % of the variation and was dominated by
SHF at all depths and $R_n$ (Table 2). PC2 explained 25% of the variation, and was
dominated by air and soil temperatures. The first three PCs explained about 86 % of the
GPP variation. For $R_{ec}$, PC1 explained 31 % of the variation and was dominated by
SWC at all depths and by precipitation (Table 3). PC2 also explained 31 % of the
variation, but was dominated by air and soil temperatures. The first three PCs explained





about 78 % of the $R_{ec}$ variation.
In autumn, PC1 explained 46 % of the NEE variation and was dominated by $T_{air}$,
SWC at depth of 10 cm, $T_{soil}$ at all depths, and $R_n$ (Table 1). PC2 explained 34 % of the
variation and was dominated by SWC at depths of 0 to 30 cm. The first two PCs
explained about 80 % of the NEE variation. For GPP, PC1 explained 33 % of the
variation and was dominated by SHF at all depths and $T_{air}$ (Table 2). PC2 explained 28 %
of the variation and was dominated by SWC and $T_{soil}$ at all depths and $R_n$. The first four
PCs explained about 85 % of the GPP variation. For $R_{ec}$, PC1 explained 36 % of the
variation and was dominated by SHF at all depths and $T_{air}$ (Table 3). PC2 explained 32%
of the variation and was dominated by SWC and $T_{soil}$ at all depths and by $R_n$. The first
three PCs explained about 82 % of the $R_{ec}$ variation.
In winter, the NEE were equal to $R_{ec}$. PC1 for NEE ($R_{ec}$) explained 39 % of the
variation and was dominated by SWC at depths of 20 to 30 cm and $T_{soil}$ at all depths
(Table 1 and Table 3). PC2 accounted for 25 % of the variation and was dominated by
SHF at a depth of 10 cm and $T_{air}$. For GPP, there was no photosynthesis during the
winter, so no data is provided in Table 2.
In summary, the dominant control factors for NEE, GPP, and $R_{ec}$ differed among the
seasons.
**4 Discussion**
**4.1 Annual and seasonal mean and diurnal variability**
Our results suggested that the sandy grassland ecosystem in China's Horqin Sandy
Land was a net $CO_2$ source, with an annual mean NEE of 48.88 $\pm$ 8.10 g C m$^{-2}$ yr$^{-1}$ in
the years for which a complete dataset was available (2015, 2016, and 2018). This result
was similar to that obtained for a semi-desert sandy grassland near Vácrátót, Hungary
(where the dominant species were *Festuca vaginata* and *Stipa capillata*), but the
Hungarian annual NEE was higher, at 131.48 g C m$^{-2}$ yr$^{-1}$ in 2001 (Balogh et al., 2005).
In contrast, many other arid and semiarid dry ecosystems with similar climate and
geographical conditions were a significant net sink for $CO_2$. For example, in the Mojave
Desert ecosystem in the United States, where the dominant species were evergreen





shrubs, drought-deciduous shrub species, and perennial grasses, the annual NEE was
$-102 \pm 67$ and $-110 \pm 70$ g C m$^{-2}$ yr$^{-1}$ in 2005 and 2006, respectively (Wohlfahrt et al.,
2008). China's Tengger Desert, where the dominant vegetation was xerophytic shrubs
planted in 1956, had annual NEE of $-13.87$ and $-23.36$ g C m$^{-2}$ yr$^{-1}$ in 2009 and 2010,
respectively (Gao et al., 2012). The southern edge of China's Mu Us desert, which is
dominated by a mixture of deciduous shrub species, had an annual NEE of $-77$ g C m$^{-}$
$^2$ yr$^{-1}$ in 2012 (Jia et al., 2014). China's Gurbantonggut Desert, which is dominated by
shrubs and grasses, had an annual NEE of $-5$ and $-40$ g C m$^{-2}$ yr$^{-1}$ in 2006 and 2007,
respectively (Liu et al., 2016a). The reason for these differences from the present study
may be differences in the carbon sequestration ability of the dominant vegetation.
Zheng et al. (2007) showed that the average carbon sequestration of terrestrial higher
plants was higher for shrubs than for herbs. The dominant vegetation of our study area
comprised annual herbs, which would have lower carbon sequestration capacity than in
a shrub-dominated ecosystem.
The sandy grassland ecosystem in the present study was a net $CO_2$ source at an annual
scale. On the one hand, this is because the dominant plants were annual plants, with a
low carbon sequestration capacity. On the other hand, the site is still recovering from
severe degradation, and has relatively low vegetation productivity (e.g., the mean
annual GPP ($351.78 \pm 20.97$ g C m$^{-2}$ yr$^{-1}$) in our study was lower in China's Mu Us
desert ($456 \pm 20.97$ g C m$^{-2}$ yr$^{-1}$) (Jia et al., 2014)), and the restoration of degraded
sandy grassland ecosystems is a long process (Li et al., 2019). Therefore, the ecosystem
has not yet reached the threshold at which it will change into a $CO_2$ sink, and it will be
necessary to study NEE for a longer period to reveal when that change occurs and the
ecosystem's long-term response to environmental and biological factors (Su et al., 2003;
Niu et al., 2018).
We believe that seasonal variation of environmental factors also explained the
seasonal differences in NEE, GPP, and R$_{ec}$ at our site. In spring, the sandy grassland
was a net $CO_2$ source in all years (Fig. 3a). Before the growing season, plants begin to
germinate, and both GPP and R$_{ec}$ increased with increasing temperature, solar radiation,



and precipitation (Niu et al., 2011; Rey et al., 2011). However, $R_{ec}$ was more responsive
than GPP to precipitation. Liu et al. (2016a) showed that precipitation before the
growing season had an important impact on NEE in arid and semiarid regions. After
the winter drought, the spring precipitation greatly promoted the respiration of soil
microbes (Zhang et al., 2016). As a result, $R_{ec}$ increased significantly. Precipitation also
promoted GPP to some extent, but the carbon uptake was relatively small during plant
germination. Therefore, the ecosystem was a net $CO_2$ source.
In summer, the sandy grassland was a $CO_2$ sink in all years (Fig. 3b). Our results
agree with previous results for the study area (Li et al., 2015), as well as with results
for a semiarid savanna in Australia (Hutley et al., 2005) and a grassland in California
(Ma et al., 2007). GPP and $R_{ec}$ increased because of the favorable temperature and
moisture conditions. However, because photosynthesis is greater than respiration
during the peak of the growing season, the ecosystem became a net $CO_2$ sink (Kemp,
1983; Liu et al., 2016a; Niu et al., 2018).
In the autumn and winter, the sandy grassland was a net $CO_2$ source in all years (Fig.
3c,d). At the end of the growing season (in autumn), annual plants began to die and
photosynthesis weakened (Fang et al., 2014). As a result, the ecosystem gradually
transformed from a carbon sink to a carbon source (Keenan et al., 2009; Kiely et al.,

2009).

At the diurnal scale, NEE in the spring, summer, and autumn showed $CO_2$ uptake
during the day (06:00-18:00), and $CO_2$ emission during the night (Fig. 4a, b, c). The
NEE increased with increasing light intensity during the day, reached its peak value
around noon, then decreased until sunset, when the ecosystem changed from net carbon
absorption to carbon release (Wagle and Kakani, 2014; Jia et al., 2014).
In winter, the sandy grassland ecosystem showed $CO_2$ emission throughout the day
(Fig. 4d). At a diurnal scale, the ecosystem showed carbon "uptake", at a level too small
to display in Fig. 4d. This phenomenon may have resulted from heating effects in the
open-path infrared gas analyzer (Burba et al., 2008). We recently created a Li-Cor LI-
8150 gas analyzer system with six long-term monitoring chambers in the footprint of



eddy covariance to test whether that hypothesis is correct.
**4.2 Impacts of the environment on NEE, GPP, and $R_{ec}$**
Our results demonstrated the important roles of the environmental factors in
regulating the direction and amount of NEE between the atmosphere and the ecosystem
in a sandy grassland in the Horqin Sandy Land. The most important environmental
factors differed among the different scales.
At the diurnal scale, NEE in the four seasons was mainly explained by the $R_n$, which
agrees with results for a study of the Mojave Desert ecosystem (Wohlfahrt et al., 2008).
Our study area was located at a relatively high latitude, which means that solar radiation
may be a limiting factor on many ecosystem processes such as GPP (Li et al., 2005;
Liang et al., 2012).
At the seasonal scale, the carbon cycle processes were affected by many
environmental factors, including soil and air temperatures, SWC, solar radiation ($R_n$
and SHF), and precipitation. Our results showed that the dominant environmental
factors that affected NEE differed among the seasons. Our PCA analysis (Tables 1, 2
and 3) showed that in the spring, the main environmental factors that affected NEE,
GPP, and $R_{ec}$ were temperature and SWC. After experiencing the winter cold and
drought, the effect of temperature and SWC on soil thawing and vegetation greenup
were greater than those of other environmental factors (Chu et al., 2013; Wolf et al.,

2016).

In summer, the most important environmental factors for NEE and GPP were solar
radiation and SHF. This result agreed with previous studies, which demonstrated that
solar radiation was the main environmental factor that affected photosynthesis during
the peak of the growing season (Saigusa et al., 2008; Hinko-Najera et al., 2017).
However, SWC was the most important factor for $R_{ec}$, and the variation of SWC was
mainly controlled by precipitation (Fig. S4b). Studies have suggested that the burst-type
precipitation could strongly stimulate $R_{ec}$ during the growing season in semiarid areas
(Hunt et al., 2002; Saetre and Stark, 2005).
In autumn, SHF and air and soil temperatures were the dominant environmental



factors for NEE, GPP, and $R_{ec}$. The ecosystem was dominated by $R_{ec}$ during the later
stages of the growing season, and studies have shown that $R_{ec}$ was strongly affected by
soil temperature (Wang et al., 2012; Niu et al., 2018), and that the changes of soil
temperature depended on SHF (Gao et al., 2010; Guo et al., 2011). Therefore, SHF and
temperature were the most important environmental factors for the autumn NEE, GPP,
and $R_{ec}$.

In winter, the annual plants had withered, so there was no GPP and the entire

ecosystem was characterized by carbon emission. Our results showed that SWC and
soil temperature were the most important factors that affected NEE, and that NEE
increased with decreasing SWC and temperature. Previous studies found that when
SWC decreases sufficiently to create water stress, it may replace temperature as the
main factor that controls soil respiration in arid and semiarid areas, and as a result, soil
respiration decreased with decreasing SWC (Wu et al., 2010; Escolar et al., 2015). Our
results were inconsistent with these previous studies. This may be due to drought, since
precipitation during the winter amounted to between 1 and 6 % of the annual
precipitation, and this would be exacerbated by strong winter winds in the Horqin
Sandy Land (Fig. S5; Wang et al., 2005; Liu et al., 2016b). The soil organic matter and
nutrients would also be lost faster when SWC decreases and the wind strengthens,
resulting in increased carbon emission (Lai, 2004; Munodawafa, 2011).

We also analyzed the relationship between annual NEE, GPP, and $R_{ec}$ in the years for

which a complete dataset was available (2015, 2016, and 2018) and the environmental
factors (Table S1). We found that the total annual precipitation was the most important
factor that limited NEE, GPP, and $R_{ec}$. NEE was negatively correlated with annual
precipitation, GPP was positively correlated with it, and the correlation between $R_{ec}$ and
precipitation was not significant. Taken together, these results indicated different
sensitivity of GPP and $R_{ec}$ to annual precipitation.

Previous studies suggested that GPP was limited by the availability of water and was

strongly correlated with total annual precipitation in arid and semiarid ecosystems
(Webb et al., 1987; Sala et al., 1988). GPP of annual herbaceous plants was especially


strongly affected by precipitation, which could change the composition and community
structure of plants, thereby affecting GPP (Nackley et al., 2014; Wang et al., 2016). Our
result was consistent with these previous studies. However, the correlation between $R_{ec}$
and precipitation was not significant in our study. This may be because of the relatively
high latitude of our study area, since $R_{ec}$ was affected by multiple environmental factors
that would be affected by latitude, such as temperature and solar radiation. However,
we must improve our understanding of the responses of the ecosystem to precipitation
and the underlying mechanisms that control whether it will be a carbon source or sink.
To accomplish this, it will be necessary to observe the ecosystem continuously for a
longer period of time.
**5 Conclusions**

Our field data indicated that the sandy grassland has functioned as a $CO_2$ source at

an annual scale, with a mean annual NEE of 48.88 $\pm$ 8.10 g C m$^{-2}$ yr$^{-1}$. At the seasonal
scale, the sandy grassland showed net $CO_2$ absorption during the summer, but net $CO_2$
release in the other seasons. At the diurnal scale, the ecosystem showed a strong single
daytime absorption peak in the spring, summer, and autumn, but strong $CO_2$ emission
at night. In winter, the ecosystem was characterized by $CO_2$ emission all day, as there
was no GPP.

At the daily scale, NEE in all four seasons was controlled by $R_n$. At the seasonal

scale, NEE was mainly affected by temperature and SWC in the spring, solar radiation
in the summer, SHF and temperature in the autumn, and SWC and temperature in the
winter. At the annual scale, the total annual precipitation was the most important factor
for NEE. Our findings demonstrated the importance of long-term, high-frequency field
monitoring in sandy land to improve our understanding of $CO_2$ cycling and its likely
responses to a changing climate. However, it will be necessary to study the NEE for a
longer period to reveal its long-term response to environmental and biological factors.

*Data availability*. In agreement with the FAIR Data standards, the data used in this

article are archived, published, and available in a dedicated repository:
http://doi.org/10.4121/uuid:35deeb02-8165-49b7-af8d-160d537ae15a.



*Competing interests.* The authors declare that they have no conflict of interest.
*Author contributions.* YQL, YYN, HBY, XYW, and YLD designed the study; YYN,
XWG, and JL performed the experiments. YYN and HBY analyzed the data. YYN
drafted the manuscript. All co-authors had a chance to review the manuscript and
contributed to discussion and interpretation of the data.
*Acknowledgements*. This research was supported by the National Key Research and
Development Program of China (2017YFA0604803 and 2016YFC0500901), the
National Natural Science Foundation of China (grants 31971466, 31560161, 31260089,
and 31400392), the Chinese Academy of Sciences "Light of West China" Program
(18JR3RA004), and the One Hundred Person Project of the Chinese Academy of
Sciences (Y551821).

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





**Figure captions**
**Fig. 1.** Locations of the Horqin Sandy Land and the Naiman station. (b) and (c) are the
covariance site at the Naiman station during the growing and dormant seasons,
respectively.

**Fig. 2.** Annual patterns of daily net ecosystem $CO_2$ exchange (NEE), gross primary
productivity (GPP), and ecosystem respiration ($R_{ec}$) from 2014 to 2018. Positive NEE
values indicate net $CO_2$ release, whereas negative values indicate net $CO_2$ uptake by the
ecosystem. Note that the initial measurements were from 15 September to 23 December
2014, so no data are available for the first part of 2014.

**Fig. 3.** Seasonal mean net ecosystem $CO_2$ exchange (NEE), gross primary productivity
(GPP), and ecosystem respiration ($R_{ec}$) from 2014 to 2018: (a) spring (March, April,
and May), (b) summer (June, July, and August), (c) autumn (September, October, and
November), and (d) winter (December, January, and February). Note that the initial
measurements were from 15 September to 23 December 2014, so no data are available
for the first part of 2014.

**Fig. 4.** Diurnal changes in mean net ecosystem $CO_2$ exchange (NEE), gross primary
productivity (GPP), and ecosystem respiration ($R_{ec}$) from 2014 to 2018: (a) spring
(March, April, and May), (b) summer (June, July, and August), (c) autumn (September,
October, and November), and (d) winter (December, January, and February). Note that
the initial measurements were from 15 September to 23 December 2014, so the spring
and summer data do not include the period before 15 September. The final
measurements were obtained on 31 December 2018, so the winter period from 2017 to
2018 was only about one-third of the usual length (i.e., it did not include data from
January and February 2019).





**Fig.1.**

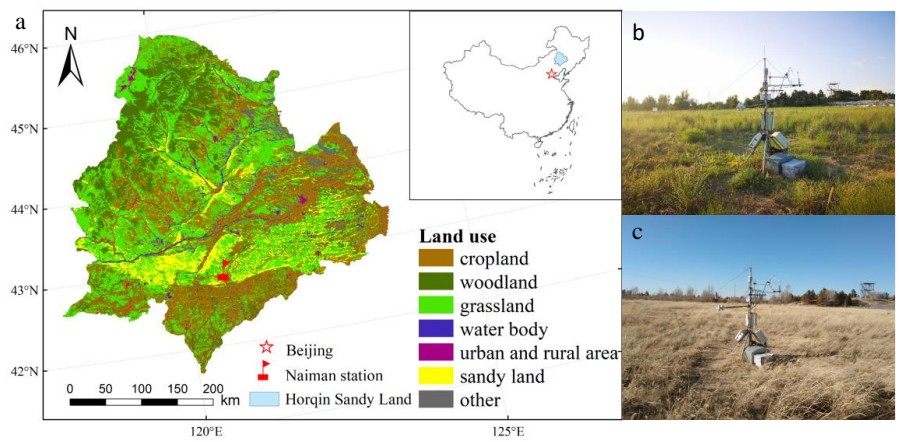


**Fig. 2.**

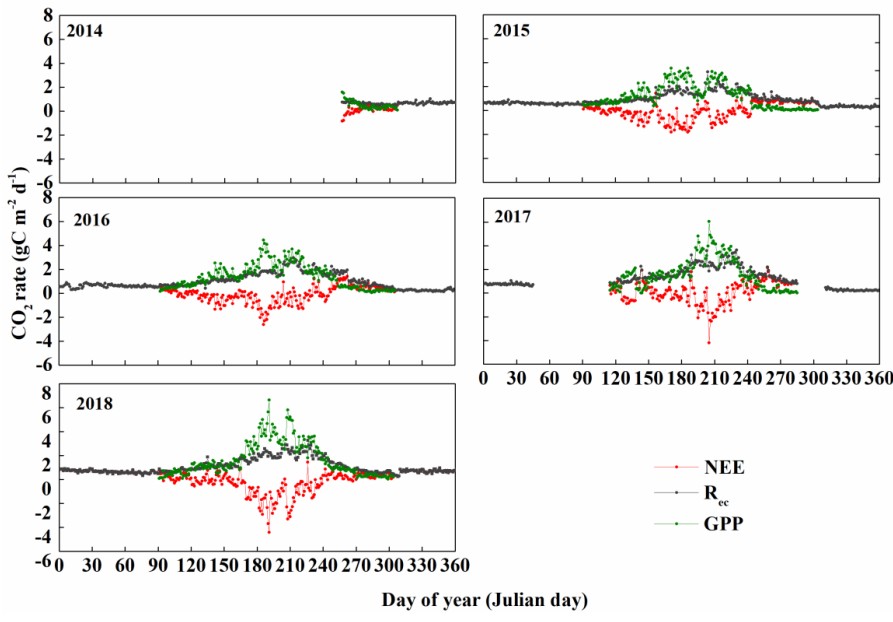







**Fig. 3.**

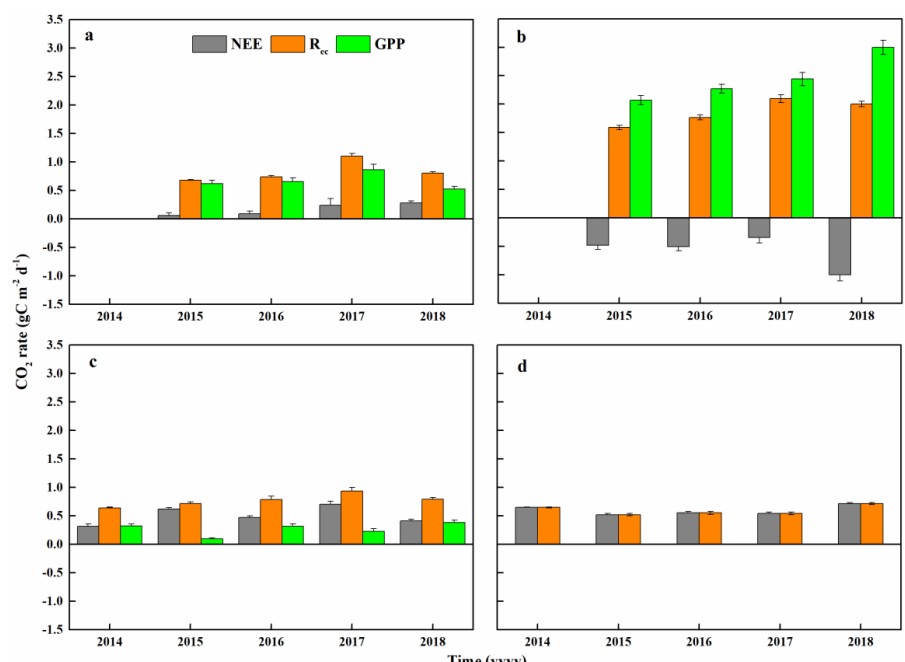







**Fig. 4.**

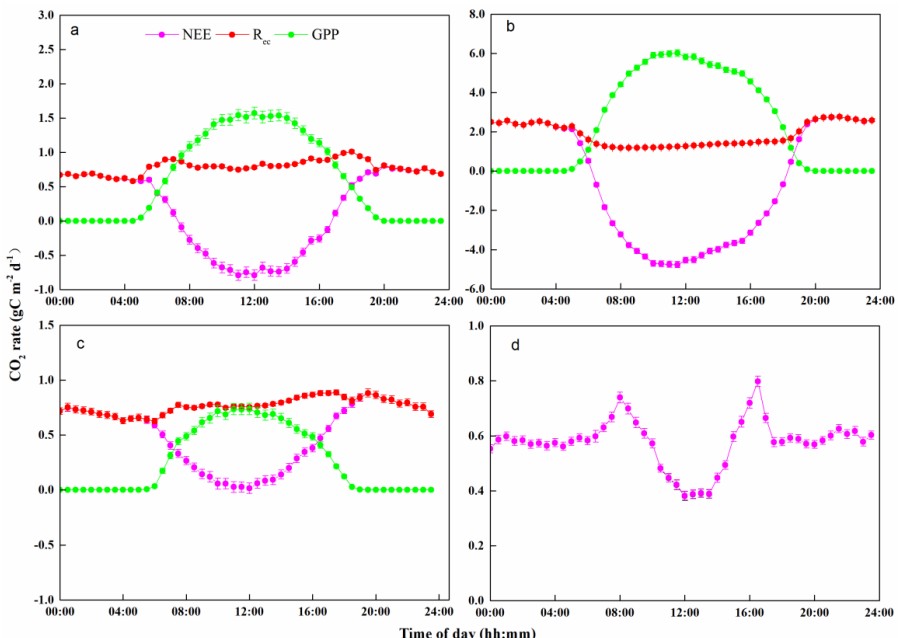







**Table 1.** Principal-components analysis (PCA) for the relationships between the net
ecosystem $CO_2$ exchange (NEE) and the environmental factors at a seasonal scale. PC,
principal component; $T_{air}$, air temperature; $R_n$, net solar radiation; SHF, soil heat flux;
SWC, soil water content; $T_{soil}$, soil temperature.

| Component[a] | Spring | | Summer | | | Autumn | | Winter | | |
|---|---|---|---|---|---|---|---|---|---|---|
| | PC1 | PC2 | PC1 | PC2 | PC3 | PC1 | PC2 | PC1 | PC2 | PC3 |
| $T_{air}$ | 0.910 | 0.188 | 0.423 | 0.861 | -0.017 | 0.873 | 0.424 | 0.333 | 0.891 | 0.071 |
| Wind speed | | | | | | | | 0.057 | 0.230 | 0.730 |
| $R_n$ | 0.739 | 0.034 | 0.781 | 0.018 | 0.032 | 0.757 | 0.418 | -0.106 | 0.320 | -0.784 |
| SHF at 5 cm | | | 0.920 | 0.198 | -0.036 | | | | | |
| SHF at 10 cm | | | 0.929 | 0.202 | 0.047 | 0.870 | -0.072 | -0.206 | 0.928 | -0.090 |
| SWC at 0-10 cm | 0.112 | 0.978 | | | | 0.238 | 0.824 | | | |
| SWC at 10-30 cm | | | | | | 0.133 | 0.845 | | | |
| SWC at 10-50 cm | 0.780 | 0.510 | | | | | | | | |
| SWC at 20-30 cm | | | | | | | | 0.868 | 0.053 | -0.144 |
| SWC at 30-40 cm | | | | | | | | 0.949 | -0.011 | 0.161 |
| SWC at 40-50 cm | | | 0.025 | -0.052 | 0.998 | | | 0.880 | -0.110 | 0.144 |
| $T_{soil}$ at 0-50 cm | 0.935 | 0.140 | -0.009 | 0.961 | -0.055 | 0.780 | 0.508 | 0.735 | 0.390 | 0.244 |
| Percent of variance | 57.227 | 25.339 | 41.639 | 29.146 | 16.735 | 46.258 | 33.520 | 39.212 | 24.720 | 16.088 |
| Cumulative | 57.227 | 82.566 | 41.639 | 70.785 | 87.521 | 46.258 | 79.778 | 39.212 | 63.933 | 80.021 |

[a] Before the PCA, SWC was divided into six depth ranges (0 to 10, 10 to 30, 10 to 50,
20 to 30, 30 to 40, and 40 to 50 cm) according to the results of a collinearity test for the
four seasons. $T_{soil}$ was divided into a single range (0 to 50 cm) according to the results
of a collinearity test for the different seasons.





**Table 2**. Principal-components analysis (PCA) for the relationships between the gross
primary productivity (GPP) and the environmental factors at the seasonal scale.
Because there was no plant photosynthesis in winter, we did not perform the PCA for
that season. PC, principal component; $T_{air}$, air temperature; $R_n$, net solar radiation; SHF,
soil heat flux; SWC, soil water content; $T_{soil}$, soil temperature.

| Component[a] | Spring | | | Summer | | | Autumn | | | |
|---|---|---|---|---|---|---|---|---|---|---|
| | PC1 | PC2 | PC3 | PC1 | PC2 | PC3 | PC1 | PC2 | PC3 | PC4 |
| $T_{air}$ | 0.849 | 0.410 | 0.108 | 0.431 | 0.849 | -0.096 | 0.736 | 0.594 | 0.178 | -0.004 |
| Wind speed | | | | | | | 0.023 | -0.001 | 0.011 | 0.980 |
| Rn | 0.511 | 0.550 | 0.136 | 0.781 | 0.027 | 0.080 | 0.557 | 0.597 | 0.115 | -0.299 |
| SHF at 5 cm | 0.138 | 0.967 | -0.059 | 0.920 | 0.173 | -0.134 | 0.928 | 0.057 | -0.091 | -0.007 |
| SHF at 10 cm | 0.191 | 0.951 | 0.032 | 0.932 | 0.185 | -0.052 | 0.943 | 0.096 | -0.063 | 0.071 |
| SWC at 0-10 cm | 0.184 | -0.006 | 0.974 | -0.073 | -0.256 | 0.890 | 0.133 | 0.758 | 0.244 | 0.007 |
| SWC at 10-50 cm | 0.836 | 0.103 | 0.416 | 0.010 | -0.017 | 0.933 | 0.000 | 0.848 | -0.071 | 0.060 |
| Precipitation | | | | | | | -0.055 | 0.146 | 0.962 | 0.008 |
| $T_{soil}$ at 0-50 cm | 0.974 | 0.126 | 0.029 | -0.003 | 0.950 | -0.170 | 0.558 | 0.685 | 0.206 | -0.131 |
| Percent of variance | 38.847 | 33.358 | 16.530 | 35.926 | 25.057 | 24.689 | 32.594 | 27.846 | 12.117 | 11.960 |
| Cumulative | 38.847 | 72.205 | 88.735 | 35.926 | 60.984 | 85.673 | 32.594 | 60.441 | 72.558 | 84.518 |

[a] Before the PCA, SWC was divided into two depth ranges (0 to 10, and 10 to 50)
according to the results of a collinearity test for the four seasons. $T_{soil}$ was divided into
a single range (0 to 50 cm) according to the results of a collinearity test for the different
seasons.





**Table 3.** Principal-components analysis (PCA) for the relationships between the ecosystem respiration ($R_{ec}$) and the environmental factors at the seasonal scale. PC, principal component; $T_{air}$, air temperature; $R_n$, net solar radiation; SHF, soil heat flux; SWC, soil water content; $T_{soil}$, soil temperature.

| Component[a] | Spring | | | Summer | | | Autumn | | | Winter | | |
|---|---|---|---|---|---|---|---|---|---|---|---|---|
| | PC1 | PC2 | PC3 | PC1 | PC2 | PC3 | PC1 | PC2 | PC3 | PC1 | PC2 | PC3 |
| $T_{air}$ | 0.844 | 0.429 | 0.040 | -0.064 | 0.912 | 0.141 | 0.726 | 0.603 | 0.179 | 0.333 | 0.891 | 0.071 |
| Wind speed | | | | 0.010 | 0.030 | 0.995 | | | | 0.057 | 0.230 | 0.730 |
| $R_n$ | 0.615 | 0.334 | -0.460 | | | | 0.554 | 0.627 | 0.118 | -0.106 | 0.320 | -0.784 |
| SHF at 5 cm | 0.162 | 0.952 | -0.197 | | | | 0.928 | 0.066 | -0.087 | | | |
| SHF at 10 cm | 0.234 | 0.929 | -0.198 | | | | 0.940 | 0.100 | -0.061 | -0.206 | 0.928 | -0.090 |
| SWC at 0-10 cm | | | | 0.811 | -0.312 | -0.091 | 0.123 | 0.754 | 0.248 | | | |
| SWC at 10-50 cm | | | | 0.884 | -0.106 | -0.013 | -0.012 | 0.838 | -0.066 | | | |
| SWC at 20-30 cm | | | | | | | | | | 0.868 | 0.053 | -0.144 |
| SWC at 20-50 cm | 0.916 | 0.061 | 0.054 | | | | | | | | | |
| SWC at 30-40 cm | | | | | | | | | | 0.949 | -0.011 | 0.161 |
| SWC at 40-50 cm | | | | | | | | | | 0.880 | -0.110 | 0.144 |
| Precipitation | 0.116 | -0.219 | 0.924 | 0.622 | 0.100 | 0.075 | -0.057 | 0.139 | 0.964 | | | |
| $T_{soil}$ at 0-50 cm | 0.946 | 0.124 | 0.050 | -0.092 | 0.942 | -0.091 | 0.549 | 0.702 | 0.207 | 0.735 | 0.390 | 0.244 |
| Percent of variance | 41.703 | 30.452 | 16.451 | 30.642 | 30.638 | 17.189 | 36.254 | 31.924 | 13.694 | 39.212 | 24.720 | 16.088 |
| Cumulative | 41.703 | 72.155 | 88.606 | 30.642 | 61.279 | 78.469 | 36.254 | 68.178 | 81.872 | 39.212 | 63.933 | 80.021 |

[a] Before the PCA, SWC was divided into six depth ranges (0 to 10, 10 to 50, 20 to 30, 20 to 50, 30 to 40, and 40 to 50 cm) according to the results of a collinearity test for the four seasons. $T_{soil}$ was divided into a single range (0 to 50 cm) according to the results of a collinearity test for the different seasons.