# Peer review of "Variations in diurnal and seasonal net ecosystem carbon dioxide exchange in a semiarid sandy grassland ecosystem in China's Horqin Sandy Land"

_Biogeosciences, 2020_

## Referee Comment (RC1) · Anonymous Referee #1 · 4 Jun 2020

General comments

Niu et al. report on 5-years of CO2 fluxes from a sandy grassland ecosystem in China's Horqin Sandy Land region. While this paper presents important information on the carbon source/sink activity of a degraded, sandy grassland system, I have concerns about the presentation and interpretation of results. Throughout the manuscript, it is unclear how some interpretations and conclusions are drawn from the presented results, and some results critical to the authors' conclusions are only found in the supplemental information. Below I address several specific concerns.

Specific comments

[Figure]

Results

While the results address an important knowledge gap on the carbon dynamics of a degraded sandy grassland, the presentation is unclear. Re-structuring the results may increase the impact and clarify of this manuscript. In its current state, the results begin with information on meteorological conditions (3.1). However, these results do not appear to be a major part of the authors' conclusions, and, from my perspective as a reader, this disrupts the flow of the manuscript. One way to re-structure the results would be to first present information on annual mean fluxes. This would address the authors' first goal: to quantify annual variation in fluxes. After presenting annual fluxes, the authors could examine seasonal then diurnal variation in fluxes. Finally, the authors could present results on meteorological conditions as possible drivers of dynamics in observed carbon fluxes.

Figure 2

In L244, the authors state "Figure 2 suggests the sandy grassland was a net $CO_2$ source." I do not see clear evidence for this in Figure 2 and it is not clear how the authors made this interpretation. Because Figure 2 depicts seasonal variation in daily $CO_2$ fluxes, it is hard to determine the sign and magnitude of annual mean carbon exchange. To make inferences about the annual source/sink activity of this system, I suggest adding a figure showing cumulative fluxes or a table depicting integrated or annual-mean fluxes. Related, the numbers listed in L244-246 show that GPP was greater than Rec, implying carbon sink behavior. However, because the reported NEE is positive, the authors conclude carbon source activity. This is very confusing and must be clarified. Please define the sign convention used for NEE.

Figure 3.

This figure is clear and provides good evidence in support of the study goals and conclusions. One suggestion would be to add another panel or figure representing annual mean fluxes, or annually integrated fluxes. The authors could then cite such a

figure as evidence of carbon source/sink behavior at the annual scale.

Figure 4.

This is a strong figure, but the interpretation in the main text is unclear. The authors report in L262-266 that NEE showed an absorption peak from 7:30 to 16:30 and that "the rest of the day was characterized by weak carbon absorption." There is no evidence for this. Before 7:30 and after 16:30, positive NEE indicates carbon emissions to the atmosphere. Please clarify. Also, I suggest adding a horizontal line to all figure at 0.0 on the y-axis. This would help the reader to quickly infer the sign carbon fluxes.

Tables 1, 2, and 3

Why is precipitation included in Tables 2 and 3 but not Table 1? One of the major study conclusions is that annual precipitation strongly regulated NEE (Section 5). However, precipitation is absent from the PCA for seasonal NEE (Table 1). The authors should explain why precipitation is not included in Table 1.

Discussion

Throughout the discussion, claims are made with no reference to evidence. For example, this happens in L379 and again in L404-405 and L425-428. These claims would be stronger if they were supported with evidence.

What I find absent in the discussion is an explanation for how drought may have influenced the interpretation of results. The authors note that the study was conducted during relatively dry years (L232-235). I appreciate that the authors considered land degradation as a possible cause of carbon source behavior. However, it would be helpful if the authors explained how interactions between land degradation and drought make it hard to attribute the observed low productivity to a single driver.

Throughout the manuscript, the definition and sign convention of NEE is unclear. This happens in the results (L244-246) and in the discussion (L415) when the authors write that NEE increased with increasing light intensity. Is this a typing error? Should this be

GPP instead of NEE?

L413: I do not see evidence of daytime CO2 uptake in autumn (Fig. 4c). Please clarify.

L448-450: The observed dependency of Rec on soil water is consistent with existing theoretical and empirical evidence that episodic rain events drive pulses of soil respiration in semiarid regions (Huxman et al., 2004; Roby et al., 2019; Sponseller, 2007).

Technical corrections

L22: please specify that these are CO2 flux measurements.

L166: Check the alignment of this text.

Supplemental material

L10: What is diurnal-scale mean value? Does this refer to the daily mean value?

Fig. S3. Panel e appears to show daily mean values for each year. Despite similar captions, panel e in Figs. S1 and S2 appear to show daily mean values averaged across years. Please clarify.

References

Huxman, T. E., Snyder, K. A., Tissue, D., Leffler, A. J., Ogle, K., Pockman, W. T., et al. (2004). Precipitation pulses and carbon fluxes in semiarid and arid ecosystems, 254–268.

Roby, M. C., Scott, R. L., Barron-Gafford, G. A., Hamerlynck, E. P., & Moore, D. J. P. (2019). Environmental and Vegetative Controls on Soil CO2 Efflux in Three Semiarid Ecosystems. Soil Systems, 3(1), 6. https://doi.org/10.3390/soilsystems3010006

Sponseller, R. A. (2007). Precipitation pulses and soil CO2 flux in a Sonoran Desert ecosystem. Global Change Biology, 13(2), 426–436.

---

## Referee Comment (RC2) · Anonymous Referee #2 · 10 Jun 2020

In this paper, the authors examined the $\sim$4.5 year record of carbon exchange, measured using eddy covariance, over a grassland site in China's Horqin Sandy land. The authors present the fluxes at diurnal, daily, monthly and yearly intervals, and use principal component analysis (PCA) to try and examine associations of the fluxes, NEE, Rec and GEP, with a whole host of hydrometeorological measurements. Their findings were limited to the associations found using the PCA with little interpretation of the PCA results. The paper, unfortunately, contained few results and insights that would be useful to the ecosystem flux community, except for perhaps the flux measurements themselves. The study lacks any hypotheses or expectations that would help guide the subsequent analysis. For example, one obvious one would be that we would expect the

seasonal to annual scale fluxes to be controlled by water availability. There are many other hypotheses and ways to analyze the data in the literature, that unfortunately, were not well reviewed either. With a lack of any physical interpretation, we instead learn things like, soil heat flux had a major effect on NEE because the authors blindly use the PCA analysis to tell us something meaningful about the grassland. This result comes from correlation between the met variables and not a mechanistic link. Some suggestions for improvement: 1. Improve the introduction and let it lead to hypotheses that you can test with the data. 2. I have included many suggested references that the authors missed. While many of these, I contributed to, they are still very relevant to this study especially because many sites in the southwest US have a similar summer monsoonal climate with similar amount of rainfall and summer temps. There are other places globally, cited in these manuscripts, that are worth looking at. These studies should prove useful to guiding your analysis and discussion and not simply presenting the data at different aggregation levels. 3. The figures only the present the data at different aggregation levels and provide little insight into what controls the seasonal to annual variation in the C fluxes. 4. I would love to see more on how water (precip, ET, soil moisture) may be controlling the warm season fluxes. 5. There is way too much reporting of data in the manuscript . For example, why is it important to know maximum and minimum values of SHF to the hundredths of W m-2?

I've also included a detailed text-specific set of comments in the attached, marked up, PDF file.

Please also note the supplement to this comment:
https://www.biogeosciences-discuss.net/bg-2020-89/bg-2020-89-RC2-supplement.pdf

——————————————————————

[Figure]

**Supplement:**

**Variations in diurnal and seasonal net ecosystem carbon dioxide exchange in a semiarid sandy grassland ecosystem in China's Horqin Sandy Land**

Yayi Niu[a,b,c,e], Yuqiang Li[a,b,c,*], Hanbo Yun[a,d,e], Xuyang Wang[a,b,c], Xiangwen Gong[a,b], Yulong Duan[a,b,c], Jing Liu[a]

[a] Northwest Institute of Eco-Environment and Resources, Chinese Academy of Sciences, Lanzhou 730000, China

[b] University of Chinese Academy of Sciences, Beijing 100049, China

[c] Naiman Desertification Research Station, Northwest Institute of Eco-Environment and Resources, Chinese Academy of Sciences, Tongliao 028300, China

[d] State Key Laboratory of Frozen Soil Engineering, Northwest Institute of Eco-Environment and Resources, Chinese Academy of Sciences, Lanzhou, Gansu 730000, China

[e] Center for Permafrost (CENPERM), Department of Geosciences and Natural Resource Management, University of Copenhagen, DK-1350 Copenhagen, Denmark

*Correspondence to*: Yuqiang Li (liyq@lzb.ac.cn)

**Abstract**

Grasslands are major terrestrial ecosystems in arid and semiarid regions, and play important roles in the regional carbon dioxide ($CO_2$) balance and cycles. Sandy grasslands are sensitive to climate change, yet the magnitudes, patterns, and environmental controls of their $CO_2$ flows are poorly understood. Here, we report the results from continuous year-round $CO_2$ observations in 5 years from a sandy grassland in China's Horqin Sandy Land. The sandy grassland was a net $CO_2$ source at an annual scale, with a mean annual net ecosystem $CO_2$ exchange (NEE) of $48.88 \pm 8.10$ g C m$^{-2}$ yr$^{-1}$ in the years for which a complete dataset was available (2015, 2016, and 2018); total annual precipitation was the most important factor for NEE. At a seasonal scale, the sandy grassland showed net $CO_2$ absorption during the summer, but net $CO_2$ release in the other seasons. The main environmental factors that affected NEE were temperature and soil water content (SWC) in the spring, soil heat flux and solar radiation in the summer, soil heat flux and temperature in autumn, and SWC and temperature in winter. At the diurnal scale, net solar radiation was the most important factor for NEE in all seasons. The sandy grassland may have been a net annual $CO_2$ source at an annual scale because the study site is recovering from degradation, thus vegetation productivity is still low. Therefore, the ecosystem has not yet transitioned to a $CO_2$ sink and long-term observations will be necessary to reveal the true source or sink intensity and its response to environmental and biological factors.

**Keywords:** Net ecosystem $CO_2$ exchange (NEE); Gross primary productivity (GPP); ecosystem respiration ($R_{ec}$); Eddy covariance; Horqin Sandy Land

**1 Introduction**

Arid and semiarid ecosystems cover 30 to 40 % of the global terrestrial surface (Poulter et al., 2014). The extent and distribution of these areas are increasing in response to factors such as climate change, changes in wildfire frequency and intensity, and changes in land use (Asner et al., 2003; Hastings et al, 2010). These ecosystems are important because they account for 30 to 35 % of terrestrial net primary productivity (Gao et al., 2012; Liu et al., 2016a) and approximately 15 % of the global soil organic

Number: 1     Author: Owner     Subject: Highlight   Date: 6/9/2020 10:02:26 AM
affected implies causation, but the analysis only shows correlation

Number: 2     Author: Owner     Subject: Highlight   Date: 6/9/2020 10:03:18 AM
Could have it been because precipitation was below normal ?

carbon pool (Lal, 2004; Liu et al., 2016a). Thus, the high potential carbon sequestration in arid and semiarid areas may have a greater impact on the future global carbon cycle than sequestration in tropical rainforest areas (Emmerich, 2003; Nosetto et al., 2006;

Poulter et al., 2014), and arid and semiarid ecosystems will have significant effects on the global carbon cycle and carbon balance (Lal, 2004). However, arid and semiarid ecosystems have received much less attention than wetter ecosystems, and the lack of high-frequency continuous observations has led to a weak understanding of their role in global terrestrial net ecosystem $CO_2$ exchange (NEE) (Baldocchi et al., 2001;

Hastings et al., 2010).

Desertification occurs in more than two-thirds of the area of arid and semiarid ecosystems (Lal, 2001). This may cause a serious imbalance in the structure and function of these ecosystems (Huenneke et al., 2002; Vest et al., 2011), especially in terms of whether the ecosystem functions as a carbon source or sink (Shachak et al.,

1998; Gang et al., 2011). Grazing exclusion is a common method used to combat desertification in the world's arid and semiarid areas (Mureithi et al., 2010; Sousa et al.,

2012). For example, Sun et al. (2015) suggested that proper exclosures promoted the recovery of degraded sandy grassland and more sustainably use sandy grassland resources.

The Horqin Sandy Land is the largest sandy land in China, and nearly 80% of the area has been desertified (Li et al., 2019). The sandy land includes multiple overlapping ecotones, including transition zones between areas with different population pressures, between semi-humid and semiarid areas, and typical agro-pastoral ecotones. The ecological environment is fragile and extremely sensitive to climate change and human activities (Bagan et al., 2010; Zhao et al., 2015). The region's sandy grassland grows on aeolian sandy soils or with sandy soils as the substrate, and is typical of the grassland vegetation that develops in sandy land. This grassland ecosystem is widespread in the

Horqin Sandy Land (Munkhdalai et al., 2007; Zhao et al., 2007). Research showed that the restoration of degraded sandy grassland can increase its productivity and carbon sequestration, and that the ecosystem can begin to act as a carbon sink (Ruiz-Jaen and

Number: 1    Author: Owner    Subject: Highlight    Date: 6/9/2020 10:06:43 AM

Drylands add up because of their vastness, but by themselves (on a per unit area basis) are not considered a potentially large area for long-term carbon sequestration.

From what has been shown, dryland C cycling contributes a lot of the interannual variability of global terrestrial C flux (fast response), but long term sequestration potential has not been borne out (slow response.

See Poulter et al. and studies that have followed that up over regions like Australia.

Number: 2    Author: Owner    Subject: Highlight    Date: 6/9/2020 10:13:24 AM

This is overlooking a large pool of work in the last 10 yrs. Some of which should certainly be mentioned and considered throughout this paper.

Please see (to name a few):

From Southwest US, which has a very similar summer monsoon climate that should be very apropos to your study.

Biederman, Joel A., et al. "CO 2 exchange and evapotranspiration across dryland ecosystems of southwestern North America." Global Change Biology 23.10 (2017): 4204-4221.

Scott, Russell L., et al. "The carbon balance pivot point of southwestern US semiarid ecosystems: Insights from the 21st century drought." Journal of Geophysical Research: Biogeosciences 120.12 (2015): 2612-2624.

Scott, Russell L., et al. "Effects of seasonal drought on net carbon dioxide exchange from a woody-plant-encroached semiarid grassland." Journal of Geophysical Research: Biogeosciences 114.G4 (2009).

Biederman, Joel A., et al. "Terrestrial carbon balance in a drier world: the effects of water availability in southwestern North America." Global Change Biology 22.5 (2016): 1867-1879.

Kurc, S. A., & Small, E. E. (2007). Soil moisture variations and ecosystem-scale fluxes of water and carbon in semiarid grassland and shrubland. Water Resources Research, 43(6).

Petrie, M. D., et al. "Grassland to shrubland state transitions enhance carbon sequestration in the northern Chihuahuan Desert." Global Change Biology 21.3 (2015): 1226-1235.
Biederman papers, Scott, Litvak, Bowling, Wagle, Aussie papers

From Australia. See papers from Beringer, Cleverly, Eamus...

Number: 3    Author: Owner    Subject: Highlight    Date: 6/9/2020 10:06:54 AM

Number: 4    Author: Owner    Subject: Highlight    Date: 6/9/2020 7:49:15 AM

what is a sandy land?

Number: 5    Author: Owner    Subject: Highlight    Date: 6/9/2020 7:50:07 AM

citation?

Aide, 2005; Zhao et al., 2016). However, other studies showed that it was a carbon source (Li et al., 2012; Niu et al., 2018). There have been relatively few long-term studies of sandy grassland at the ecosystem level, so we do not yet fully understand the characteristic of NEE and its components, gross primary productivity (GPP) and ecosystem respiration ($R_{ec}$), at an ecosystem scale, particularly for sandy grassland protected by grazing exclosures, and more data are needed, particularly for semiarid sandy land (Barrett, 1968; Czobel et al., 2012).

Precipitation is one of the factors that most strongly affects NEE in arid and semiarid areas. It affects NEE mainly through its effects on GPP and $R_{ec}$ (Lal, 2004; Dasci et al, 2010; Hastings et al, 2010; Liu et al., 2016b). Previous research showed that reduced precipitation caused a continental-scale reduction in GPP, with concurrent decreases of $R_{ec}$ (Ciais et al., 2005). In contrast, other studies found that GPP was more sensitive than respiration to precipitation (Thomas et al., 2010; Shi et al., 2014). For instance, Delgado-Balbuena et al. (2019) have shown that GPP was more than twice as sensitive as $R_{ec}$ to precipitation in a semiarid grassland. The precipitation in the Horqin Sandy Land is low, with high temporal and spatial variation, and other water resources such as groundwater are small (Niu et al., 2015). Moreover, we found no reports on the response of ecosystem-scale NEE and its components to precipitation, and the response mechanisms are uncertain in the sandy grassland. Therefore, long-term data are required to fully understand how changes in annual precipitation influence the annual NEE, GPP, and $R_{ec}$ of sandy grassland.

In this paper, we present the results from continuous (14 September 2014 to 31 December 2018) *in situ* monitoring of $CO_2$ dynamics in the Horqin Sandy Land's sandy grassland using the eddy covariance technique, and quantified the temporal variation of NEE and the factors that control it. We had the following goals: (1) To quantify the diurnal, seasonal, and annual variation in NEE, GPP, and $R_{ec}$. (2) To identify the environmental factors that controlled NEE and its components at different temporal scales, and the possible underlying mechanisms in the sandy grassland. (3) To determine how annual precipitation affects the annual ecosystem NEE, GPP, and $R_{ec}$.

**Page: 4**

**2 Materials and methods**

**2.1 Experimental site**

Our study was conducted in a sandy grassland in the southern part of the Horqin Sandy Land, Inner Mongolia, China, at the Naiman Desertification Research Station of the Chinese Academy of Sciences (42 °55′ N, 120 °42′ E) (Fig. 1). The terrain is flat, and it evolved from reclamation of sandy grassland for agriculture to severe desertification, after which cultivation was abandoned and grazing exclosures were established to allow natural recovery of the vegetation, starting in 1985 (Zhao et al., 2007). Thus, the grassland had been recovering naturally for nearly 30 years when our study began. At an elevation of 377 m a.s.l., the study area has a continental semiarid monsoon temperate climate regime. The mean annual temperature is 6.8 ℃, with mean monthly temperatures ranging from -9.63 ℃ in January to 24.58 ℃ in July. Average annual precipitation is approximately 360 mm, with 70% of the precipitation occurring during the growing season, between June and August. Annual mean potential evaporation is approximately 1973 mm. The annual frost-free period is 130 to 150 days. The zonal soil is a sandy chestnut soil, but most of the soil has been degraded by a combination of climate change and anthropogenic activity (unsustainable grazing or agriculture) into an aeolian sandy soil under the action of wind erosion (Zhao et al., 2007), with coarse sand, fine sand, and clay-silt contents of 92.7, 3.3, and 4.0 % in the topsoil to a depth of 20 cm. The contents of soil organic carbon and total nitrogen were 1.27 and 0.21 g kg$^{-1}$, respectively. Vegetation cover in the study area ranged from 50 to 70 %. The dominant plant species were annual herbs, including *Artemisia scoparia, Setaria viridis, Salsola collina*, and *Corispermum hyssopifolium* (Niu et al., 2018).

**2.2 Eddy covariance observations**

An eddy covariance flux tower (2.0 m high) was installed at the center of the observation field (Fig. 1b, c). We have continuously monitored $CO_2$, water, and heat fluxes at the tower using the eddy covariance system since late 2014. The fetch from all directions was more than 200 m. Calculations with a footprint model indicated that the fetch was well within the desired flux footprint (Schmid, 1994; Xu and Baldocchi,

Number: 1          Author: Owner          Subject: Highlight   Date: 6/9/2020 10:22:02 AM

Given the dominant controls of water in semiarid regions and that these P values and summer temps are very similar to the monsoon region in N. America, there is even more reason to bring in some of these results (found in the papers I listed above) in the Introduction and discussion.

Number: 2          Author: Owner          Subject: Highlight   Date: 6/9/2020 8:02:04 AM

?

Number: 3          Author: Owner          Subject: Highlight   Date: 6/9/2020 10:26:49 AM

You would want the fetch (region upwind of the flux tower that consists of "homogeneous" vegetation type and cover) to be greater than the flux footprint (See Schmid, H. P. "Experimental design for flux measurements: matching scales of observations and fluxes." Agricultural and Forest Meteorology 87.2-3 (1997): 179-200.)

2004). The eddy covariance system consisted of an LI-7500 infrared gas analyzer (Li-Cor Inc., Lincoln, NE, USA), with a precision of 0.01 μmol $m^{-2}$ $s^{-1}$ and an accuracy within 1 % of the reading for measurements at 30-min mean intervals, and a CSAT-3 three-dimensional ultrasonic anemometer (Campbell Scientific, Inc., Logan, UT, USA), with a precision of 0.1 ℃ and an accuracy of 1 % for the readings at 30-min mean intervals. Raw 10-Hz data were recorded by a CR3000 datalogger (Campbell Scientific). The operation, calibration, and maintenance of the eddy covariance system followed the manufacturers' standard procedures. The LI-7500 was calibrated every 6 months for $CO_2$, water vapor, and dew point values using calibration gases and dew point generator measurements supported by the China Land–Atmosphere Coordinated Observation System (Yun et al., 2018). We cleaned the mirror of the LI-7500 every 15 days to maintain the automatic gain control value below its threshold (55 to 65). All of the instruments were powered by solar panels connected to a battery.

**2.3 Micrometeorological measurements**

Along with the flux measurements obtained by the eddy covariance equipment, we measured standard meteorological and soil parameters continuously with an array of sensors. A propeller anemometer was installed at the top of the meteorological tower to measure the wind speed and direction. Net radiation ($R_n$, W $m^{-2}$) was measured by a four-component radiometer (CNR-1, Kipp and Zonen, Delft, the Netherlands) installed at 1 m above the ground. The air temperature ($T_{air}$, ℃) and relative humidity (%) instruments (HMP45C, Vaisala Inc., Helsinki, Finland) were mounted at 2 m above the ground to measure the $T_{air}$, relative humidity, and atmospheric pressure (kPa). Precipitation (mm) measurements were obtained from a meteorological station 400 m from the study site.

We installed five CS109 temperature probes (Campbell Scientific) and five CS616 moisture probes (Campbell Scientific) in the soil at depths of 10, 20, 30, 40, and 50 cm to measure soil temperature ($T_{soil}$, ℃) and soil water content (SWC, %). Two self-calibrating HFP01 soil heat flux (SHF, W $m^{-2}$) sensors (Hukseflux, Delft, the Netherlands) were buried 5 and 10 cm below the ground to obtain the SHF data. All of

the environmental parameters were measured simultaneously with the eddy covariance measurements, and all data were recorded as 30-min mean values with a CR3000

datalogger.

**2.4 Data quality and gap-filling method**

We used the EddyPro 6.2.0 software (https://www.licor.com/env/products/eddy_covariance/software.html) to process the

10-Hz raw eddy covariance data. Processing included spike removal, lag correction, secondary coordinate rotation, Webb–Pearman–Leuning correction, and sonic virtual temperature conversion (Webb et al., 1980). We used the data processing method of Lee et al. (2004) to process the 30-min mean raw flux measurements to ensure their quality.

Processed data were further corrected for weather effects and sensor uncertainty using the following procedure: (1) We removed data gathered during precipitation events, power failures, and sensor maintenance or malfunction. (2) We excluded unrealistic

$CO_2$ flux data (values outside the range of –2.0 to 2.0 mg $CO_2$ $m^{-2}$ $s^{-1}$). (3) We rejected the data during $1$able atmospheric conditions at nighttime (friction velocity ($u^*$) < 0.1

m $s^{-1}$). Based on the $R_n$, NEE was classified as the daytime $NEE_{day}$ ($R_n \geq 1$ W $m^{-2}$) or the night-time $NEE_{night}$ ($R_n < 1$ W $m^{-2}$). This screening resulted in the rejection of 20 to

30 % of the flux data, depending on the period.

We used several strategies to compensate for missing data. We used linear interpolation to fill gaps that were shorter than 2 h. For longer gaps, we handled the gap in the $NEE_{day}$ using the mean diurnal variation with a 7-day window (Falge et al., 2001), and handled the gap in the $NEE_{night}$ using a temperature-dependent exponential model (Lloyd and Taylor, 1994):

$NEE_{night} = R_0 \exp (b \, T_{10})$      Eq (1)

where $R_0$ is the base ecosystem respiration rate when the soil temperature is 0 ℃, b an empirically determined coefficient, and $T_{10}$ is the soil temperature at a depth of 10

cm. Daytime ecosystem respiration can be estimated by extrapolation from the parameterization derived from Eq. (1). We did not attempt to fill in gaps longer than 7

days, and treated those gaps as missing data. Gross primary productivity (GPP) was

Number: 1    Author: Owner    Subject: Highlight    Date: 6/9/2020 10:27:56 AM
Atmospheric stability depends on more than other factors besides u*

Number: 2    Author: Owner    Subject: Highlight    Date: 6/9/2020 10:28:26 AM
How were these parameters obtained and how often? A moving window of ~1 week should be used.  See for example, Reichstein et al. 2005

obtained as follows:

$GPP = NEE - R_{ec}$ Eq (2)

[revised manuscript text omitted]

Number: 1    Author: Owner    Subject: Highlight    Date: 6/9/2020 10:29:31 AM
All of this extreme detailed reporting of numeral results is unnecessary. Use figures and tables to report only what is necessary to your analysis.

Number: 2    Author: Owner    Subject: Highlight    Date: 6/9/2020 10:36:53 AM
This could be why the NEE was positive, not just grassland recovery trajectory as mentioned in the abstract and discussion.

See Scott et al. 2010, 2015

Number: 3    Author: Owner    Subject: Highlight    Date: 6/9/2020 10:38:16 AM
Too much reporting of the values already shown in the figures and really the figures all convey essentially the same data, just at different time scales.

Number: 4    Author: Owner    Subject: Highlight    Date: 6/9/2020 10:39:32 AM
Here and elsewhere non-significant figures are being reported. Do we really believe that precip is accurate down to the nearest 10th of a mm, fluxes down to the 100th of a g?

ranged from 255.99 g C m$^{-1}$ yr$^{-1}$ in 2015 to 355.77 g C m$^{-2}$ yr$^{-1}$ in 2018 and R$_{ec}$ ranged from 319.04 g C m$^{-2}$ yr$^{-1}$ in 2015 to 390.85 g C m$^{-2}$ yr$^{-1}$ in 2018. The eddy covariance tower was set up in mid-August 2014, and the instrument was allowed to stabilize for 1 month before we began collecting data, from 15 September to 23 December 2014; during that period, we measured a cumulative carbon release of 46.67 g C m$^{-2}$, with cumulative GPP and R$_{ec}$ of 24.80 and 71.47 g C m$^{-2}$, respectively. From 15 February to 26 April 2017 and from 14 October to 6 November 2017, approximately 3 months of data were missing due to instrument maintenance and calibration, and the cumulative NEE, GPP, and R$_{ec}$ were 63.33, 273.67 and 337.00 g C m$^{-2}$ for the remaining 9 months of the year. Note that the periods covered by the data are therefore not identical.

We also observed clear seasonal variations in the total seasonal NEE, GPP, and R$_{ec}$ (Fig. 3) and their diurnal cycles (Fig. 4). In spring, the sandy grassland was an atmospheric $CO_2$ source in all years, with NEE, GPP, and R$_{ec}$ averaging $0.15 \pm 0.05$, $0.63 \pm 0.03$, and $0.78 \pm 0.03$ g C m$^{-2}$ d$^{-1}$, respectively (Fig. 3a). The diurnal NEE cycle was characterized by a single absorption peak from around 07:30 to around 16:30 (Fig. 4a). Note that although all times in China are reported as the Beijing time, the study site was not sufficiently far east of Beijing for this to affect the physiological meaning of these times. The rest of the day was characterized by weak carbon absorption. The average diurnal GPP was also characterized by a single peak from around 05:00 to around 19:30, and the diurnal R$_{ec}$ was characterized by an approximately horizontal line at about 0.78 g C m$^{-2}$ d$^{-1}$. The maximum and minimum average diurnal values for NEE were 0.78 (20:00) and −0.81 (12:00) g C m$^{-2}$ d$^{-1}$, respectively, versus 1.57 (12:00) and 0.58 (4:30) g C m$^{-2}$ d$^{-1}$ for GPP and 1.01 (18:00) and 0.05 (17:30) g C m$^{-2}$ d$^{-1}$ for R$_{ec}$.

In summer, the sandy grassland was a $CO_2$ sink in all years, with NEE, GPP, and R$_{ec}$ averaging $−0.58 \pm 0.05$, $2.44 \pm 0.05$, and $1.86 \pm 0.03$ g C m$^{-2}$ d$^{-1}$, respectively (Fig. 3b). The diurnal cycles of NEE and GPP were also characterized by a single peak, between 05:00 and 19:30 (Fig. 4b), and the ecosystem $CO_2$ uptake reached its peak from around 10:30 to 12:00. The diurnal R$_{ec}$ pattern was similar to the spring, but at a higher level (about 1.86 g C m$^{-2}$ d$^{-1}$). The maximum and minimum diurnal NEE averaged 2.67

Number: 1          Author: Owner          Subject: Highlight   Date: 6/9/2020 10:40:29 AM
Round off to the nearest 1.

Number: 2          Author: Owner          Subject: Highlight   Date: 6/9/2020 8:15:58 AM
Unusual...the thing should work right out of the box. Do you really mean this?

(21:30) and −4.60 (11:30) g C m$^{-2}$ d$^{-1}$, respectively, versus 6.02 (11:30) and 0.09 (19:30)

g C m$^{-2}$ d$^{-1}$ for GPP and 2.77 (21:30) and 1.19 (8:00) g C m$^{-2}$ d$^{-1}$ for $R_{ec}$.

In autumn, the sandy grassland was a net source of atmospheric $CO_2$ in all years, with NEE, GPP, and $R_{ec}$ averaging 0.50 ±0.02, 0.27 ±0.02, and 0.76 ±0.02 g C m$^{-2}$

d$^{-1}$, respectively (Fig. 3c). The diurnal dynamics of NEE, GPP, and $R_{ec}$ in autumn (Fig.

4c) were similar to those in spring (Fig. 4a), but the magnitudes of NEE and GPP in autumn were lower than in the spring. The diurnal $R_{ec}$ was similar to the spring, at about

0.76 g C m$^{-2}$ d$^{-1}$. The maximum and minimum average diurnal NEE were 0.88 (19:00)

and 0.02 (11:30) g C m$^{-2}$ d$^{-1}$, respectively, versus 0.89 (17:30) and 0.63 (4:00) g C m$^{-2}$

d$^{-1}$ for GPP and 0.74 (12:00) and 0.01 (5:00) g C m$^{-2}$ d$^{-1}$ for $R_{ec}$.

In winter, the grassland ecosystem functioned as a net $CO_2$ source in all years, with an average seasonal NEE of 0.58 ±0.01 g C m$^{-2}$ d$^{-1}$ (Fig. 3d). It should also be noted that since the investigation started on 14 September 2014 and ended on 31 December

2018, the 2017 to 2018 winter was only about one-third of the usual length (i.e., it did not include data from January and February 2019). The diurnal dynamics of the winter

NEE differed from the other seasons (Fig. 4d), with a minimum release value of 0.38 g

C m$^{-2}$ d$^{-1}$, and with two emission peaks: at 0.81 g C m$^{-2}$ d$^{-1}$ (08:00) and 0.89 g C m$^{-2}$

d$^{-1}$ (16:30).

**1.3 Response of NEE, GPP and $R_{ec}$ to changes in environmental factors**

We analyzed the effects of environmental factors on NEE and its components at different temporal scales. The analysis methods for the diurnal scale (Pearson's *r* and

PCA) were the same as the methods used at a seasonal scale, so to avoid repetition, we have only described the relationship between the seasonal-scale NEE and its components and the associated environmental factors. At the diurnal scale, $R_n$ was the main factor that affected NEE, GPP and $R_{ec}$ in all four seasons (data not shown). NEE

was significantly negatively correlated with $R_n$, whereas GPP and $R_{ec}$ were positively correlated with $R_n$, indicating that the ecosystem's carbon sequestration capacity increased with increasing $R_n$.

Seasonal-scale NEE, GPP, and $R_{ec}$ were significantly correlated with many

Number: 1     Author: Owner     Subject: Highlight   Date: 6/9/2020 10:43:05 AM
I'd like to see some plots like GEP or Reco vs. ppt (yearly or monthly or seasonal), et, or soil moisture. Water availability should be a dominant control. If it isn't, you need to tell us why.

Number: 2     Author: Owner     Subject: Highlight   Date: 6/9/2020 10:44:40 AM
This is not an in-depth analysis.  We already know that the diurnal scale is not only controlled, but also defined, by the fluctuations in energy input.

Number: 3     Author: Owner     Subject: Highlight   Date: 6/9/2020 8:24:43 AM

Number: 4     Author: Owner     Subject: Highlight   Date: 6/9/2020 10:45:18 AM
Is this across seasons or within season?

environmental factors (Table S1). We found extremely weak and non-significant relationships between NEE, GPP, and $R_{ec}$ and two climate variables (relative humidity and atmospheric pressure), so we excluded those variables from our subsequent analysis. NEE was negatively correlated with most environmental factors in all seasons except the autumn, when most correlations were positive, and GPP and $R_{ec}$ were positively correlated with most environmental factors (Table S1).

In spring, $T_{soil}$ at all depths, and $T_{air}$, $R_n$, and SWC at all depths were negatively correlated with NEE (Table S1). In the PCA, principal component 1 (PC1) explained 57 % of the NEE variation (Table 1), and was dominated by temperature ($T_{soil}$ at all depths and $T_{air}$). PC2 explained about 25 % of the NEE variation, and was dominated by SWC at depths of 0 to 10 cm. The first two PCs explained about 82 % of the NEE variation. GPP was positively correlated with most environmental factors. PC1 explained 39 % of the GPP variation, and temperature and SWC at depths of 10 to 50 cm were the dominant factors (Table 2). PC2 explained about 33 % of GPP variation, and was dominated by SHF at all depths. The first three PCs explained about 89 % of the GPP variation. $R_{ec}$ was positively correlated with all environmental factors except for wind speed. PC1 explained 42 % of the $R_{ec}$ variation (Table 3) and was dominated by SWC at depth of 20 to 50 cm, $T_{soil}$ at all depths, $T_{air}$, and $R_n$. PC2 explained about 30% of the $R_{ec}$ variation, and was dominated by SHF at all depths. The first three PCs explained about 89 % of the $R_{ec}$ variation.

In summer, PC1 explained 42 % of the NEE variation, and was dominated by SHF at all depths and $R_n$ (Table 1). PC2 explained 29% of the NEE variation, and was dominated by air and soil temperatures. The first three PCs explained about 88 % of the NEE variation. For GPP, PC1 explained 36 % of the variation and was dominated by SHF at all depths and $R_n$ (Table 2). PC2 explained 25% of the variation, and was dominated by air and soil temperatures. The first three PCs explained about 86 % of the GPP variation. For $R_{ec}$, PC1 explained 31 % of the variation and was dominated by SWC at all depths and by precipitation (Table 3). PC2 also explained 31 % of the variation, but was dominated by air and soil temperatures. The first three PCs explained

**Number: 1**      Author: Owner      Subject: Highlight    Date: 6/9/2020 10:45:27 AM
Should use VPD

**Number: 2**      Author: Owner      Subject: Highlight    Date: 6/9/2020 10:46:14 AM
This type of analysis isn't getting to the heart of the matter. T is related to the seasonality or phenology at the site because it matches the timing of rainfall input (covariation), but it is water, not energy, which is a first order control on carbon cycling in these regions.

**Number: 3**      Author: Owner      Subject: Highlight    Date: 6/9/2020 10:46:41 AM
What's the relationship between GPP and ground heat flux?

**Number: 4**      Author: Owner      Subject: Highlight    Date: 6/9/2020 8:33:30 AM
third PC isn't discussed

**Number: 5**      Author: Owner      Subject: Highlight    Date: 6/9/2020 8:34:16 AM
same two comments above apply here

**Number: 6**      Author: Owner      Subject: Highlight    Date: 6/9/2020 10:49:07 AM
This is a great example of why this analysis is not useful. The diurnal cycle is a dominated by energy. You need to look beyond this first order constraint on the diurnal variability to what is controlling the seasonal strength or weakness of GPP and Rec. Again, if that isn't water, then why not?

**Number: 7**      Author: Owner      Subject: Highlight    Date: 6/9/2020 10:51:05 AM
These results of within season empirical PC analysis provides no useful insight into the controls on grassland C flux that the rest of the community could find useful.

about 78 % of the $R_{ec}$ variation.

In autumn, PC1 explained 46 % of the NEE variation and was dominated by $T_{air}$,

SWC at depth of 10 cm, $T_{soil}$ at all depths, and $R_n$ (Table 1). PC2 explained 34 % of the variation and was dominated by SWC at depths of 0 to 30 cm. The first two PCs explained about 80 % of the NEE variation. For GPP, PC1 explained 33 % of the variation and was dominated by SHF at all depths and $T_{air}$ (Table 2). PC2 explained 28 %

of the variation and was dominated by SWC and $T_{soil}$ at all depths and $R_n$. The first four

PCs explained about 85 % of the GPP variation. For $R_{ec}$, PC1 explained 36 % of the variation and was dominated by SHF at all depths and $T_{air}$ (Table 3). PC2 explained 32%

of the variation and was dominated by SWC and $T_{soil}$ at all depths and by $R_n$. The first three PCs explained about 82 % of the $R_{ec}$ variation.

In winter, the NEE were equal to $R_{ec}$. PC1 for NEE ($R_{ec}$) explained 39 % of the variation and was dominated by SWC at depths of 20 to 30 cm and $T_{soil}$ at all depths (Table 1 and Table 3). PC2 accounted for 25 % of the variation and was dominated by

SHF at a depth of 10 cm and $T_{air}$. For GPP, there was no photosynthesis during the winter, so no data is provided in Table 2.

In summary, the dominant control factors for NEE, GPP, and $R_{ec}$ differed among the seasons.

**4 Discussion**

**4.1 Annual and seasonal mean and diurnal variability**

Our results suggested that the sandy grassland ecosystem in China's Horqin Sandy

Land was a net $CO_2$ source, with an annual mean NEE of 48.88 $\pm$ 8.10 g C m$^{-2}$ yr$^{-1}$ in the years for which a complete dataset was available (2015, 2016, and 2018). This result was similar to that obtained for a semi-desert sandy grassland near Vácrátót, Hungary (where the dominant species were *Festuca vaginata* and *Stipa capillata*), but the

Hungarian annual NEE was higher, at 131.48 g C m$^{-2}$ yr$^{-1}$ in 2001 (Balogh et al., 2005).

In contrast, many other arid and semiarid dry ecosystems with similar climate and geographical conditions were a significant net sink for $CO_2$. For example, in the Mojave

Desert ecosystem in the United States, where the dominant species were evergreen

Number: 1     Author: Owner     Subject: Highlight   Date: 6/9/2020 8:40:44 AM
This is not a publishable result

Number: 2     Author: Owner     Subject: Highlight   Date: 6/9/2020 10:52:34 AM
Recommend comparing your results to comparable ecosystems in comparable climates.  Missing Scott et al. , Biederman 2018, Pietrie et al. Studies from Australia or S. Africa.

shrubs, drought-deciduous shrub species, and perennial grasses, the annual NEE was $-102 \pm 67$ and $-110 \pm 70$ g C m$^{-2}$ yr$^{-1}$ in 2005 and 2006, respectively (Wohlfahrt et al., 2008). China's Tengger Desert, where the dominant vegetation was xerophytic shrubs planted in 1956, had annual NEE of $-13.87$ and $-23.36$ g C m$^{-2}$ yr$^{-1}$ in 2009 and 2010, respectively (Gao et al., 2012). The southern edge of China's Mu Us desert, which is dominated by a mixture of deciduous shrub species, had an annual NEE of $-77$ g C m$^{-2}$ yr$^{-1}$ in 2012 (Jia et al., 2014). China's Gurbantonggut Desert, which is dominated by shrubs and grasses, had an annual NEE of $-5$ and $-40$ g C m$^{-2}$ yr$^{-1}$ in 2006 and 2007, respectively (Liu et al., 2016a). The reason for these differences from the present study may be differences in the carbon sequestration ability of the dominant vegetation. Zheng et al. (2007) showed that the average carbon sequestration of terrestrial higher plants was higher for shrubs than for herbs. The dominant vegetation of our study area comprised annual herbs, which would have lower carbon sequestration capacity than in a shrub-dominated ecosystem.

The sandy grassland ecosystem in the present study was a net $CO_2$ source at an annual scale. On the one hand, this is because the dominant plants were annual plants, with a low carbon sequestration capacity. On the other hand, the site is still recovering from severe degradation, and has relatively low vegetation productivity (e.g., the mean annual GPP ($351.78 \pm 20.97$ g C m$^{-2}$ yr$^{-1}$) in our study was lower in China's Mu Us desert ($456 \pm 20.97$ g C m$^{-2}$ yr$^{-1}$) (Jia et al., 2014)), and the restoration of degraded sandy grassland ecosystems is a long process (Li et al., 2019). Therefore, the ecosystem has not yet reached the threshold at which it will change into a $CO_2$ sink, and it will be necessary to study NEE for a longer period to reveal when that change occurs and the ecosystem's long-term response to environmental and biological factors (Su et al., 2003; Niu et al., 2018).

We believe that seasonal variation of environmental factors also explained the seasonal differences in NEE, GPP, and $R_{ec}$ at our site. In spring, the sandy grassland was a net $CO_2$ source in all years (Fig. 3a). Before the growing season, plants begin to germinate, and both GPP and $R_{ec}$ increased with increasing temperature, solar radiation,

Number: 1    Author: Owner    Subject: Highlight    Date: 6/9/2020 10:53:43 AM
Certainly not a proven result and there are contradictory, site-specific results. See Scott et al. 2015, Kurc and Small 2007, Pietrie et al. ...

Number: 2    Author: Owner    Subject: Highlight    Date: 6/9/2020 10:54:47 AM
What about below-average precipitation?  Again, our expectation is that water is the dominant control so this should be examined.

Number: 3    Author: Owner    Subject: Highlight    Date: 6/9/2020 10:55:34 AM
This doesn't convey any useful information.

[revised manuscript text omitted]

In autumn, SHF and air and soil temperatures were the dominant environmental
* * *
**T** Number: 1        Author: Owner       Subject: Highlight   Date: 6/9/2020 9:46:55 AM

This is the main result?
* * *
**T** Number: 2        Author: Owner       Subject: Highlight   Date: 6/9/2020 10:56:00 AM

The daily scale is by definition regulated by radiation input.
* * *
**T** Number: 3        Author: Owner       Subject: Highlight   Date: 6/9/2020 9:48:54 AM

I would expect to see WATER here.
* * *
Author: Owner     Subject: Sticky Note       Date: 6/9/2020 10:59:40 AM

Because your study relies upon simple empirical correlation analysis to tell you what's going on without any input from what is already know from the science, your result here is overly simplistic with no links to physical processes. For example, why would SHF ever be relevant to GPP?

factors for NEE, GPP, and $R_{ec}$. The ecosystem was dominated by $R_{ec}$ during the later stages of the growing season, and studies have shown that $R_{ec}$ was strongly affected by soil temperature (Wang et al., 2012; Niu et al., 2018), and that the changes of soil temperature depended on SHF (Gao et al., 2010; Guo et al., 2011). Therefore, SHF and temperature were the most important environmental factors for the autumn NEE, GPP, and $R_{ec}$.

In winter, the annual plants had withered, so there was no GPP and the entire ecosystem was characterized by carbon emission. Our results showed that SWC and soil temperature were the most important factors that affected NEE, and that NEE

increased with decreasing SWC and temperature. Previous studies found that when

SWC decreases sufficiently to create water stress, it may replace temperature as the main factor that controls soil respiration in arid and semiarid areas, and as a result, soil respiration decreased with decreasing SWC (Wu et al., 2010; Escolar et al., 2015). Our results were inconsistent with these previous studies. This may be due to drought, since precipitation during the winter amounted to between 1 and 6 % of the annual precipitation, and this would be exacerbated by strong winter winds in the Horqin

Sandy Land (Fig. S5; Wang et al., 2005; Liu et al., 2016b). The soil organic matter and nutrients would also be lost faster when SWC decreases and the wind strengthens, resulting in increased carbon emission (Lai, 2004; Munodawafa, 2011).

We also analyzed the relationship between annual NEE, GPP, and $R_{ec}$ in the years for which a complete dataset was available (2015, 2016, and 2018) and the environmental factors (Table S1). We found that the total annual precipitation was the most important factor that limited NEE, GPP, and $R_{ec}$. NEE was negatively correlated with annual precipitation, GPP was positively correlated with it, and the correlation between $R_{ec}$ and precipitation was not significant. Taken together, these results indicated different sensitivity of GPP and $R_{ec}$ to annual precipitation.

Previous studies suggested that GPP was limited by the availability of water and was strongly correlated with total annual precipitation in arid and semiarid ecosystems (Webb et al., 1987; Sala et al., 1988). GPP of annual herbaceous plants was especially

strongly affected by precipitation, which could change the composition and community structure of plants, thereby affecting GPP (Nackley et al., 2014; Wang et al., 2016). Our result was consistent with these previous studies. However, the correlation between $R_{ec}$

and precipitation was not significant in our study. This may be because of the relatively high latitude of our study area, since $R_{ec}$ was affected by multiple environmental factors that would be affected by latitude, such as temperature and solar radiation. However, we must improve our understanding of the responses of the ecosystem to precipitation and the underlying mechanisms that control whether it will be a carbon source or sink.

To accomplish this, it will be necessary to observe the ecosystem continuously for a longer period of time.

**5 Conclusions**

Our field data indicated that the sandy grassland has functioned as a $CO_2$ source at an annual scale, with a mean annual NEE of 48.88 $\pm$ 8.10 g C m$^{-2}$ yr$^{-1}$. At the seasonal scale, the sandy grassland showed net $CO_2$ absorption during the summer, but net $CO_2$

release in the other seasons. At the diurnal scale, the ecosystem showed a strong single daytime absorption peak in the spring, summer, and autumn, but strong $CO_2$ emission at night. In winter, the ecosystem was characterized by $CO_2$ emission all day, as there was no GPP.

At the daily scale, NEE in all four seasons was controlled by $R_n$. At the seasonal scale, NEE was mainly affected by temperature and SWC in the spring, solar radiation in the summer, SHF and temperature in the autumn, and SWC and temperature in the winter. At the annual scale, the total annual precipitation was the most important factor for NEE. Our findings demonstrated the importance of long-term, high-frequency field monitoring in sandy land to improve our understanding of $CO_2$ cycling and its likely responses to a changing climate. However, it will be necessary to study the NEE for a longer period to reveal its long-term response to environmental and biological factors.

*Data availability*. In agreement with the FAIR Data standards, the data used in this article are archived, published, and available in a dedicated repository:

http://doi.org/10.4121/uuid:35deeb02-8165-49b7-af8d-160d537ae15a.

Number: 1     Author: Owner     Subject: Highlight    Date: 6/9/2020 11:00:48 AM
Where is this result shown?

Number: 2     Author: Owner     Subject: Highlight    Date: 6/9/2020 9:58:08 AM
This dataset appears to have all the data to remake the figures.  What would really be useful is to know where the community can access the 30 min met and flux data? This "raw" data is exactly what others could use to generate comparisons and include with future studies.  I strongly suggest this data is shared in a flux archive like ChinaFlux/Fluxnet.  This will only increase the use of this data and benefit this studies authors.

[revised manuscript text omitted]

---

## Author Comment (AC1) · 4 Jul 2020

Reviewer 1

RE: Submission of the revised manuscript (No. bg-2020-89): Variations in diurnal and seasonal net ecosystem carbon dioxide exchange in a semiarid sandy grassland ecosystem in China's Horqin Sandy Land.

Dear Reviewer#1: Thank you very much for your assistance in the review of our manuscript. We have revised the manuscript carefully according to your comments. We have also had this revised manuscript edited by Mr. Geoffrey Hart

(ghart@videotron.ca/geoff@geoff-hart.com), an English science editor with nearly 30 years of experience, to ensure that the quality of the language will be acceptable. Please contact him if necessary to confirm that he has performed this work or if you have any questions about the nature of the work that he has done. Our detailed responses to comments are presented in the remainder of this letter. All of revisions have been highlighted in red in the revision.

General comments:

Niu et al. report on 5-years of $CO_2$ fluxes from a sandy grassland ecosystem in China's Horqin Sandy Land region. While this paper presents important information on the carbon source/sink activity of a degraded, sandy grassland system, I have concerns about the presentation and interpretation of results. Throughout the manuscript, it is unclear how some interpretations and conclusions are drawn from the presented results, and some results critical to the authors' conclusions are only found in the supplemental information. Below I address several specific concerns:

1. Results. While the results address an important knowledge gap on the carbon dynamics of a degraded sandy grassland, the presentation is unclear. Re-structuring the results may increase the impact and clarify of this manuscript. In its current state, the results begin with information on meteorological conditions (3.1). However, these results do not appear to be a major part of the authors' conclusions, and, from my perspective as a reader, this disrupts the flow of the manuscript. One way to re-structure the results would be to first present information on annual mean fluxes. This would address the authors' first goal: to quantify annual variation in fluxes. After presenting annual fluxes, the authors could examine seasonal then diurnal variation in fluxes. Finally, the authors could present results on meteorological conditions as possible drivers of dynamics in observed carbon fluxes.

Although re-structuring the Results section has some advantages, the meteorological conditions (3.1) provide essential context for understanding our results, as they are

the primary factors that drive CO2 fluxes in the sandy grassland. Therefore, we have retained the original section 3.1, but focus our analysis on the environmental factors that are drivers of the observed dynamics of the carbon fluxes (lines 264-294 in the revision). We then present the annual mean fluxes (lines 296-310 in the revision), then examine the seasonal and diurnal variation of the fluxes (lines 311-341 in the revision). Finally, we analyze the responses of the CO2 fluxes to changes in meteorological conditions as possible drivers of the observed carbon fluxes (lines 343-376 in the revision).

2. Figure 2. In L244, the authors state "Figure 2 suggests the sandy grassland was a net CO2 source." I do not see clear evidence for this in Figure 2 and it is not clear how the authors made this interpretation. Because Figure 2 depicts seasonal variation in daily CO2 fluxes, it is hard to determine the sign and magnitude of annual mean carbon exchange. To make inferences about the annual source/sink activity of this system, I suggest adding a figure showing cumulative fluxes or a table depicting integrated or annual-mean fluxes. Related, the numbers listed in L244-246 show that GPP was greater than Rec, implying carbon sink behavior. However, because the reported NEE is positive, the authors conclude carbon source activity. This is very confusing and must be clarified. Please define the sign convention used for NEE.

We have added Figure 3f to present the annual cumulative NEE, GPP, and Rec and to show the net source results more clearly (lines 296-300, Fig. 3f in the revision). We apologize for typing incorrect values for GPP and Rec, which incorrectly caused GPP to be greater than Rec. We have revised the values of the GPP and Rec and have checked all other numbers throughout the revision to prevent other errors (lines 298-299 in the revision). In response to your comments, we have defined the sign convention we used for NEE (lines 246-248 in the revision).

3. Figure 3. This figure is clear and provides good evidence in support of the study goals and conclusions. One suggestion would be to add another panel or figure representing annual mean fluxes, or annually integrated fluxes. The authors could then cite such a figure as evidence of carbon source/sink behavior at the annual scale.

We have added another panel to present the annual mean fluxes (Figure 4 in the revision).

4. Figure 4. This is a strong figure, but the interpretation in the main text is unclear. The authors report in L262-266 that NEE showed an absorption peak from 7:30 to 16:30 and that "the rest of the day was characterized by weak carbon absorption." There is no evidence for this. Before 7:30 and after 16:30, positive NEE indicates carbon emissions to the atmosphere. Please clarify. Also, I suggest adding a horizontal line to all figure at 0.0 on the y-axis. This would help the reader to quickly infer the sign carbon fluxes.

We have revised the description to clarify our meaning (lines 314-315 in the revision). We have also added a horizontal line to all graphs at 0.0 on the y-axis (Fig. 5 in the revision).

5. Tables 1, 2, and 3. Why is precipitation included in Tables 2 and 3 but not Table 1? One of the major study conclusions is that annual precipitation strongly regulated NEE (Section 5). However, precipitation is absent from the PCA for seasonal NEE (Table 1). The authors should explain why precipitation is not included in Table 1.

In response to Reviewer #2, we have removed the PCA results for seasonal NEE, GPP, and Rec, and focused on the impact of precipitation and soil water content on the $CO_2$ flux at seasonal and annual scales, because precipitation is the factor that most strongly affects the $CO_2$ flux in arid and semiarid regions (lines 343-376, 395-403, 449-504 in the revision).

6. Discussion. Throughout the discussion, claims are made with no reference to evidence. For example, this happens in L379 and again in L404-405 and L425-428. These claims would be stronger if they were supported with evidence.

We have added references to support our claims in the Discussion. These are highlighted in red in the revision.

7. What I find absent in the discussion is an explanation for how drought may have influenced the interpretation of results. The authors note that the study was conducted during relatively dry years (L232-235). I appreciate that the authors considered land degradation as a possible cause of carbon source behavior. However, it would be helpful if the authors explained how interactions between land degradation and drought make it hard to attribute the observed low productivity to a single driver.

We have added an analyses of the relationship between annual precipitation and the NEE, GPP, and Rec in the Results (Lines 343-350 and Fig. 6 in the revision), and have explained how the precipitation affected the NEE, GPP, and Rec in the Discussion (Lines 449-461 in the revision). We have also noted (lines 406-409 in the revision) that although we did not quantify the degree of degradation of the study site, our results suggest that the site has not yet recovered sufficiently to become a net sink.

8. Throughout the manuscript, the definition and sign convention of NEE is unclear. This happens in the results (L244-246) and in the discussion (L415) when the authors write that NEE increased with increasing light intensity. Is this a typing error? Should this be GPP instead of NEE?

We have revised the values of GPP and Rec and checked throughout the revision to ensure that they are correct (lines 298-299 in the revision). We have defined the sign convention used for NEE (lines 246-248 in the revision) and have revised the description in the Discussion to agree with this convention (lines 429-431 in the revision).

9. L413: I do not see evidence of daytime $CO_2$ uptake in autumn (Fig. 4c). Please clarify.

We have revised the description to correct this error (lines 433-434 in the revision).

10. L448-450: The observed dependency of Rec on soil water is consistent with existing theoretical and empirical evidence that episodic rain events drive pulses of soil respiration in semiarid regions (Huxman et al., 2004; Roby et al., 2019; Sponseller,

2007).

Thank you for bringing these papers to our attention. We have revised the description to include a citation of these papers (lines 498-500 in the revision).

Technical corrections

11. L22: please specify that these are CO2 flux measurements.

We have added that these are CO2 flux measurements (line 22 in the revision).

12. L166: Check the alignment of this text.

We have revised the alignment of the text (line 218 in the revision).

Supplemental material

13. L10: What is diurnal-scale mean value? Does this refer to the daily mean value? Fig. S3. Panel e appears to show daily mean values for each year. Despite similar captions, panel e in Figs. S1 and S2 appear to show daily mean values averaged across years. Please clarify.

We have revised Fig. S1 (e) and Fig. S2 (e) to show the daily mean values for each year in order to more intuitively display the variations in these environmental factors during the whole study period (lines 10-11, and 15 in the supplement).

Thanks for your efforts to improve our manuscript. We hope that our replies and the resulting changes will be satisfactory, but we will be happy to work with you to resolve any remaining issues.

Sincerely,

Yuqiang Li, Ph.D

Northwest Institute of Eco-Environment and Resources Chinese Academy of Sciences

320 Donggang West Road, Lanzhou, 730000, China

Phone/Fax: 86-931-496-7219

E-mail: liyq@lzb.ac.cn

Please also note the supplement to this comment:
https://www.biogeosciences-discuss.net/bg-2020-89/bg-2020-89-AC1-supplement.pdf
* * *
[Figure]

[Figure]

**Fig. 1.** Seasonal and inter-annual variation in the daily average NEE, GPP and Rec from (a-e) 2014 to 2018. (f) Annual cumulative NEE, GPP and Rec from 2014 to 2018.

[Figure]

**Fig. 2.** Seasonal mean NEE, GPP and Rec from 2014 to 2018: (a) spring, (b) summer (c) autumn, and (d) winter.

NEE ── $R_{ec}$ ── GPP

**a**

**b**

**c**

**d**

CO$_2$ rate (μmol m$^{-2}$ s$^{-1}$)

CO$_2$ rate (μmol m$^{-2}$ s$^{-1}$)

Time of day (hh:mm)

**Fig. 3.** Diurnal changes in mean NEE, GPP and Rec from 2014 to 2018: (a) spring, (b) summer (c) autumn, and (d) winter.

[Figure]

---

## Author Comment (AC2) · 4 Jul 2020

Reviewer 2

RE: Submission of the revised manuscript (No. bg-2020-89): Variations in diurnal and seasonal net ecosystem carbon dioxide exchange in a semiarid sandy grassland ecosystem in China's Horqin Sandy Land.

Dear Reviewer#2: Thank you very much for your assistance in the review of our manuscript. We have revised the manuscript carefully according to your comments. We have also had this revised manuscript edited by Mr. Geoffrey Hart

(ghart@videotron.ca/geoff@geoff-hart.com), an English science editor with nearly 30 years of experience, to ensure that the quality of the language will be acceptable. Please contact him if necessary to confirm that he has performed this work or if you have any questions about the nature of the work that he has done.

Our detailed responses to comments are presented in the remainder of this letter. All of revisions have been highlighted in red in the revision.

General comments:

In this paper, the authors examined the 4.5 year record of carbon exchange, measured using eddy covariance, over a grassland site in China's Horqin Sandy land. The authors present the fluxes at diurnal, daily, monthly and yearly intervals, and use principal component analysis (PCA) to try and examine associations of the fluxes, NEE, Rec and GEP, with a whole host of hydrometeorological measurements. Their findings were limited to the associations found using the PCA with little interpretation of the PCA results. The paper, unfortunately, contained few results and insights that would be useful to the ecosystem flux community, except for perhaps the flux measurements themselves. The study lacks any hypotheses or expectations that would help guide the subsequent analysis. For example, one obvious one would be that we would expect the seasonal to annual scale fluxes to be controlled by water availability. There are many other hypotheses and ways to analyze the data in the literature, that unfortunately, were not well reviewed either. With a lack of any physical interpretation, we instead learn things like, soil heat flux had a major effect on NEE because the authors blindly use the PCA analysis to tell us something meaningful about the grassland. This result comes from correlation between the met variables and not a mechanistic link. Some suggestions for improvement:

1. Improve the introduction and let it lead to hypotheses that you can test with the data.

We have revised the Introduction to include a specific statement of our research hypotheses and goals (lines 139-151 in the revision). Based on your comment, we have

also eliminated the PCA analysis and focused on the relationships between environmental factors and our results. We believe that our results are important enough to publish because so little data exists for our study area. We have noted this in lines 127-129 in the revision.

2. I have included many suggested references that the authors missed. While many of these, I contributed to, they are still very relevant to this study especially because many sites in the southwest US have a similar summer monsoonal climate with similar amount of rainfall and summer temps. There are other places globally, cited in these manuscripts, that are worth looking at. These studies should prove useful to guiding your analysis and discussion and not simply presenting the data at different aggregation levels.

Thank you for your suggested references. We have carefully read the literature that you recommended, and have added citations for the relationships between precipitation, soil water content, and $CO_2$ flux in different seasons based on these papers (lines 97-134, 139-151, 343-376, 395-403, 449-504 and Fig. 6-8 in the revision).

3. The figures only the present the data at different aggregation levels and provide little insight into what controls the seasonal to annual variation in the C fluxes.

We have added analyses and a discussion of the effects of precipitation and soil moisture on the seasonal and annual $CO_2$ flux (lines 343-376, 395-403, 449-504 and Fig. 6-8 in the revision).

4. I would love to see more on how water (precip, ET, soil moisture) may be controlling the warm season fluxes.

We have added an analysis of the impacts of precipitation and soil moisture on fluxes during the warm seasons (lines 357-358, 368-373, 476-494, Fig. 7-8 in the revision).

5. There is way too much reporting of data in the manuscript. For example, why is it important to know maximum and minimum values of SHF to the hundredths of W m-2?

We have removed the least important data and retained only the most important data that explain the fluxes (lines 264-271, 285-294 in the revision).

I've also included a detailed text-specific set of comments in the attached, marked up, PDF file.

We have made the changes you suggested, subject to revision by our English editor.

Page 2:

Number1: affected implies causation, but the analysis only shows correlation.

We have revised the description (lines 26-27 in the revision).

Number2: Could have it been because precipitation was below normal?

Precipitation was below normal, so you are correct that this may explain why the sandy grassland was a net $CO_2$ source at an annual scale (lines 26-27, 395-403, 449-461 in the revision).

Page 3:

Number 1: Drylands add up because of their vastness, but by themselves (on a per unit area basis) are not considered a potentially large area for long-term carbon sequestration. From what has been shown, dryland C cycling contributes a lot of the interannual variability of global terrestrial C flux (fast response), but long term sequestration potential has not been borne out (slow response. See Poulter et al. and studies that have followed that up over regions like Australia.

We have revised our description to account for the paper by Poulter et al. and other studies in arid and semiarid areas (lines 47-49 in the revision).

Number 2 and 3: This is overlooking a large pool of work in the last 10 yrs. Some of which should certainly be mentioned and considered throughout this paper. Please see (to name a few): From Southwest US, which has a very similar summer monsoon

climate that should be very apropos to your study.

Biederman, Joel A., et al. "CO2 exchange and evapotranspiration across dryland ecosystems of southwestern North America." Global Change Biology 23.10 (2017): 4204-4221.

Scott, Russell L., et al. "The carbon balance pivot point of southwestern US semi-arid ecosystems: Insights from the 21st century drought." Journal of Geophysical Research: Biogeosciences 120.12 (2015): 2612-2624.

Scott, Russell L., et al. "Effects of seasonal drought on net carbon dioxide exchange from a woody-plant-encroached semiarid grassland." Journal of Geophysical Research: Biogeosciences 114.G4 (2009).

Biederman, Joel A., et al. "Terrestrial carbon balance in a drier world: the effects of water availability in southwestern North America." Global Change Biology 22.5 (2016): 1867-1879.

Kurc, S. A., & Small, E. E. (2007). Soil moisture variations and ecosystem‐scale fluxes of water and carbon in semiarid grassland and shrubland. Water Resources Research, 43(6).

Petrie, M. D., et al. "Grassland to shrubland state transitions enhance carbon sequestration in the northern Chihuahuan Desert." Global Change Biology 21.3 (2015): 1226-1235.

Biederman papers, Scott, Litvak, Bowling, Wagle, Aussie papers From Australia. See papers from Beringer, Cleverly, Eamus...

Thank you for directing our attention to these references. We have carefully read these papers and have cited the most relevant ones (highlighted in red in the revision). We have added a summary of some of the studies that have been carried out in arid and semiarid regions according to your comments (lines 51-67 in the revision).

Number 4: what is a sandy land?

We have added a definition of the meaning of "sandy land" (lines 78-80 in the revision).

Number 5: citation?

We have added a citation (lines 86 in the revision).

Page 4:

Number 1: can't tell from the reference list which papers these are because there are no years in the listing.

We have added the years in the listings and checked throughout the revision (lines 724-726 in the revision).

Number 2: citations to Scott et al. 2015, also Biederman et al. 2015.

We have cited both papers (lines 97-98 in the revision).

Number 3: Australian papers by cleverly, beriinger.

We have cited both papers (lines 98-100 in the revision).

Number 4 and 5: how is this in contrast with "concurrent decreases"? see Scott et al., Biederman et al. 2015

We have removed our description of the sensitivity of GPP and Rec to precipitation and have revised this to focus on the amount of precipitation in terms of its impact on ecological processes in the context of our hypotheses (lines 98-105 in the revision).

Number 6: of the "sandy grassland" in this region or grasslands underlain by sandy soils more globally?

We have revised this as "of the sandy grassland" in this region (lines 129 in the revision).

Number 7: Given these pretty goals limited to mainly data reporting. Ecosystem flux

science has moved beyond this type of paper and into data interpretation and contextualization. It would be better to put these results into context of what is known already about the seasonal to interannual sensitivity elsewhere. This could be done either by bringing in these expectation learned from previous studies and testing those, or by pulling in data from other sites or syntheses like Fluxnet 2015.

First, we note that although "ecosystem flux science has moved beyond this type of paper", it is still important for researchers to provide basic data for regions that have not yet been described well, or at all, in the literature. Although your suggestions are valuable to direct future research, it would take a long time to obtain and collate these data, and there is little research on CO2 fluxes in degraded sandy grassland ecosystems, so the impact mechanisms are not clear. We have therefore used the existing data to obtain preliminary results for CO2 fluxes and identify the dominant environmental factors in the sandy grassland of our study area. This will provide an empirical basis for future theoretical research at multiple sites.

Page: 5

Number 1: Given the dominant controls of water in semiarid regions and that these P values and summer temps are very similar to the monsoon region in N. America, there is even more reason to bring in some of these results (found in the papers I listed above) in the Introduction and discussion.

We have added some description of the dominant controls of precipitation in the Introduction and Discussion (lines 97-127, 139-151, 449-504 in the revision).

Number 2: zonal soil ?

We have clarified that this is the dominant regional soil type (lines 167 in the revision).

Number 3: You would want the fetch (region upwind of the flux tower that consists of "homogeneous" vegetation type and cover) to be greater than the flux footprint (See Schmid, H. P. "Experimental design for flux measurements: matching scales of observations and fluxes." Agriculturaland Forest Meteorology 87.2-3 (1997): 179-200.)

We have clarified that the fetch is greater than the footprint and cited the Schmid paper (lines 180-182 in the revision).

Page: 7

Number 1: Atmospheric stability depends on more than other factors besides u*.

We have used u* to reflect insufficient turbulent mixing at night, which is a general method (Reichstein et al., 2005; Scott et al., 2009). We have revised the description accordingly (lines 226-229 in the revision).

Number 2: How were these parameters obtained and how often? A moving window of ∼1 week should be used. See for example, Reichstein et al.2005.

We have clarified that these parameters were obtained using the SPSS software with a 7-day window (lines 235-237 in the revision).

Page: 8

Number 1: These should be used to estimate G, the soil heat flux at z=0.

According to your comments and the actual situation in our study area, we determined that precipitation was the dominant factor that influenced the flux, so we have focused our analysis and descriptions on precipitation. The other indicators were removed.

Page: 9

Number 1: All of this extreme detailed reporting of numeral results is unnecessary. Use figures and tables to report only what is necessary to your analysis.

We have removed the unnecessary data descriptions and have retained the data most relevant to our analysis (lines 264-271, 285-294 in the revision).

Number 2: This could be why the NEE was positive, not just grassland recovery trajectory as mentioned in the abstract and discussion See Scott et al. 2010, 2015.

We analyzed the relationship between precipitation and CO2 flux according to your comment, and found that the below-normal precipitation may be one explanation for why the sandy grassland was a net CO2 source at an annual scale (lines 26-27, 395-403, 449-461 in the revision).

Number 3: Too much reporting of the values already shown in the figures and really the figures all convey essentially the same data, just at different time scales.

We have removed the values that are already shown in the figures in the revision.

Number 4: Here and elsewhere non-significant figures are being reported. Do we really believe that precip is accurate down to the nearest 10th of a mm, fluxes down to the 100th of a g?

We have rounded all values to the nearest integer in the revision.

Page: 10

Number 1: Round off to the nearest 1.

We have modified these values throughout the revision.

Number 2: Unusual...the thing should work right out of the box. Do you really mean this?

What we wanted to express was that we conducted a 1-month pre-experiment to test the stability of the instrument, and did not use the data collected during this period in our study. In order to avoid ambiguity, we have deleted this description.

Page: 11

Number 1: I'd like to see some plots like GEP or Reco vs. ppt (yearly or monthly or seasonal), et, or soil moisture. Water availability should be a dominant control. If it isn't, you need to tell us why.

We have added graphs of the relationships between NEE, GPP, Rec, and the precipitation and soil water content at different scales (yearly, monthly, and daily) (lines 343-376, 395-403, 449-504 and Fig. 6-8 in the revision).

Number 2: This is not an in-depth analysis. We already know that the diurnal scale is not only controlled, but also defined, by the fluctuations in energy input.

As we mentioned above, precipitation and soil water content were the main factors that influenced the fluxes and that we focused our descriptions on these factors rather than the energy factors. Specifically, we added an analysis of the relationships between NEE, GPP, and Rec and the soil temperature and soil water content at a daily scale (lines 362-376, and Fig. 8 in the revision).

Page: 12

Number 1: Should use VPD.

As we mentioned above, precipitation and temperature were the main factors that influenced the fluxes, so we have focused on them and removed our description of the effects of relative humidity and atmospheric pressure.

Number 2: This type of analysis isn't getting to the heart of the matter. T is related to the seasonality or phenology at the site because it matches the timing of rainfall input (covariation), but it is water, not energy, which is a first order control on carbon cycling in these regions.

We agree with you suggestion that water is the most important factor and may be the first-order control on carbon cycling in our study region, so we analyzed the relationship between the $CO_2$ flux and the precipitation and soil water content (lines 343-376, 395-403, 449-504 and Fig. 6-8 in the revision).

Number 3: What's the relationship between GPP and ground heat flux?

As we mentioned above, we have removed our discussion of the ground heat flux because it was not the main impact factor in our study region.

[Figure]

Number 4: third PC isn't discussed.

We have removed the PC analysis in the revision, because PCA analysis only shows the relationships between the variables and does not represent a mechanistic (causal) link. Because precipitation was the main influencing factor, we focused our analysis on precipitation and the resulting effects on soil water (lines 343-376, 395-403, 449-504 and Fig. 6-8 in the revision).

Number 5: same two comments above apply here.

As we mentioned above, we have removed the PC analysis in the revision.

Number 6: This is a great example of why this analysis is not useful. The diurnal cycle is a dominated by energy. You need to look beyond this first order constraint on the diurnal variability to what is controlling the seasonal strength or weakness of GPP and Rec. Again, if that isn't water, then why not?

We have removed the PC analysis and energy factors according to your comments, and have focused our analysis on the relationships between CO2 flux and the precipitation and soil water content (lines 343-376, 395-403, 449-504 and Fig. 6-8 in the revision). These results of within season empirical PC analysis provides no useful insight into the controls on grassland C flux that the rest of the community could find useful. As we mentioned above, we have deleted the PC analysis.

Page: 13

Number 1: This is not a publishable result.

We have deleted the result.

Number 2: Recommend comparing your results to comparable ecosystems in comparable climates. Missing Scott et al., Biederman 2018, Pietrie et al. Studies from Australia or S. Africa.

We have added comparisons with these references (lines 381-388 in the revision).

Page: 14

Number 1: Certainly not a proven result and there are contradictory, site-specific re-sults. See Scott et al. 2015, Kurc and Small 2007, Pietrie et al.

We have deleted the description and revised this to focus on how carbon sequestration of the ecosystem decreased with decreasing annual precipitation (lines 395-403 in the revision). Number 2: What about below-average precipitation? Again, our expectation is that water is the dominant control so this should be examined.

As we mentioned above, we analyzed the relationship between precipitation and the $CO_2$ fluxes according to your comment, and showed that the below-normal precipita-tion may be one reason why the sandy grassland was a net $CO_2$ source at an annual scale (lines 26-27, 395-403, 449-461 in the revision).

Number 3: This doesn't convey any useful information.

We have deleted the description in the revision.

Page: 16

Number 1: This is the main result?

Yes, this is one of the main results.

Number 2: The daily scale is by definition regulated by radiation input.

As we mentioned above, we have deleted energy factors because they were not the dominant factors in the semiarid area.

Number 3: I would expect to see WATER, Because your study relies upon simple empirical correlation analysis to tell you what's going on without any input from what is already know from the science, your result here is overly simplistic with no links to physical processes. For example, why would SHF ever be relevant to GPP?

We have deleted the PC analysis and added a description of how water affected the

CO2 fluxes (lines 465-504 in the revision).

Page: 18

Number 1: Where is this result shown?

We have added the result (lines 343-350 and Fig 6 in the revision).

Number 2: This dataset appears to have all the data to remake the figures. What would really be useful is to know where the community can access the 30 min met and flux data? This "raw" data is exactly what others could use to generate comparisons and include with future studies. I strongly suggest this data is shared in a flux archive like ChinaFlux/Fluxnet. This will only increase the use of this data and benefit this studies authors.

Once our results are published, we will share our 30-min data in a flux archive such as ChinaFlux/Fluxnet to support comparisons and include this data in future studies.

Page: 35

Number 1: Labels are too small on all axes of all figures.

We have modified the text on the axes of all figures.

Page: 37

Number 1: The convention for these types of diurnal plots is to use micromol CO2 m-2 sec-1

We have revised the diurnal plots to use $\mu$mol CO2 m-2 s-1.

Thank you for your comments and for your work to improve the quality of our paper. We hope that with these changes, it will now be acceptable for publication, but we will be happy to work with you to resolve any remaining issues.

Sincerely,

[Figure]

Yuqiang Li, Ph.D

Northwest Institute of Eco-Environment and Resources Chinese Academy of Sciences

320 Donggang West Road, Lanzhou, 730000, China

Phone/Fax: 86-931-496-7219

E-mail: liyq@lzb.ac.cn

Please also note the supplement to this comment:
https://www.biogeosciences-discuss.net/bg-2020-89/bg-2020-89-AC2-supplement.pdf

―――――――――――――――――――――

[Figure]

**Fig. 1.** Changes in soil water content (SWC) at depths of 10, 20, 30, 40, and 50 cm that resulted from precipitation events in spring, summer, and autumn.

[Figure]

**Fig. 2.** Relationship between annual precipitation and NEE, GPP and Rec for the years with a complete dataset (2015, 2016, and 2018).

$$GPP = 0.98\ PPT + 2.20$$
$$R^2 = 0.70$$
$$p < 0.001$$

$$R_{ec} = 0.51\ PPT + 17.55$$
$$R^2 = 0.69$$
$$p < 0.001$$

$$NEE = -0.48\ PPT + 15.35$$
$$R^2 = 0.53$$
$$p < 0.001$$

[Figure]

**Fig. 3.** Relationship between monthly NEE, GPP and Rec and the corresponding monthly precipitation in spring, summer, and autumn.

[Figure]

Spring   Summer   Autumn

**a**
$R^2 = 0.50$  slope = -0.04
$p < 0.001$
$R^2 = 0.07$ slope = -0.01
$p < 0.001$
$R^2 = 0.07$  slope = 0.01, $p < 0.001$

**b**
$R^2 = 0.02$  slope = -0.06
$p < 0.001$
$p > 0.05$
$R^2 = 0.03$  slope = 0.04
$p < 0.001$

**c**
$R^2 = 0.34$
slope = -0.18
$p < 0.001$
$R^2 = 0.04$
slope = -0.01
$p < 0.001$
$R^2 = 0.07$  slope = -0.03, $p < 0.001$

**d**
$R^2 = 0.64$ slope = 0.02
$p < 0.001$
$R^2 = 0.22$ slope = 0.09
$p < 0.001$
$R^2 = 0.60$ slope = 0.04
$p < 0.001$

**e**
$R^2 = 0.04$ slope = 0.04
$p < 0.001$
$R^2 = 0.08$ slope = 0.10
$p < 0.001$
$R^2 = 0.03$  slope = 0.04
$p < 0.001$

**f**
$R^2 = 0.43$
slope = 0.10
$p < 0.001$
$R^2 = 0.10$
slope = 0.11
$p < 0.001$
$R^2 = 0.23$
slope = 0.14
$p < 0.001$

**g**
$R^2 = 0.59$  slope = 0.07
$p < 0.001$
$R^2 = 0.19$  slope = 0.19
$p < 0.001$
$R^2 = 0.33$  slope = 0.04
$p < 0.001$

**h**
$R^2 = 0.07$ slope = 0.13
$p < 0.001$
$R^2 = 0.03$  slope = 0.13
$p < 0.001$
$R^2 = 0.03$  slope = 0.06
$p < 0.001$

**i**
$R^2 = 0.21$
slope = 0.21
$p < 0.001$
$R^2 = 0.07$
slope = 0.20
$p < 0.001$
$R^2 = 0.49$
slope = 0.17
$p < 0.001$

NEE (g C m$^{-2}$ d$^{-1}$)

R$_{ec}$ (g C m$^{-2}$ d$^{-1}$)

GPP (g C m$^{-2}$ d$^{-1}$)

T$_{soil}$ (℃)

0 to 10 cm SWC (%)

10 to 50 cm SWC (%)

**Fig. 4.** Relationships between daily NEE, GPP and Rec and the average soil temperature (Tsoil) and soil water content (SWC) in spring, summer, and autumn.

**Supplement:**

*Supplemental material*

**Variations in diurnal and seasonal net ecosystem carbon dioxide exchange in a semiarid sandy grassland ecosystem in China's Horqin Sandy Land**

Yayi Niu et al.

*Correspondence to*: Yuqiang Li (liyq@lzb.ac.cn)

[Figure]

**Fig. S1 |** Air temperature (T$_{air}$) measured 2 m above the ground surface: (a), (b), (c), and (d) are half–hourly mean values in spring, summer, autumn, and winter, respectively; (e) Seasonal and inter-annual variation in daily average T$_{air}$ from late 2014 to late 2018.

[Figure]

**Fig. S2 |** Net radiation ($R_n$) measured 2 m above the ground surface: (a), (b), (c), and

(d) are half‒hourly mean values in spring, summer, autumn, and winter, respectively;

(e) Seasonal and inter-annual variation in daily average $R_n$ from late 2014 to late 2018.

[Figure]

17

**Fig. S3** | Soil heat flux (SHF) at depths of 5 cm and 10 cm: (a), (b), (c), and (d) are

half‑hourly mean values in spring, summer, autumn, and winter, respectively; (e)

Seasonal and inter-annual variation in daily average SHF at depths of 5 cm and 10 cm

from late 2014 to late 2018.

[Figure]

23

24  **Fig. S4 | (a)** Seasonal and inter-annual variation in daily average soil temperature ($T_{soil}$)

25  and (b) soil water content (SWC) at depths of 10, 20, 30, 40, and 50 cm, and seasonal

26  and inter-annual variation in daily precipitation from late 2014 to late 2018.

[Figure]

27

28 **Fig. S5 |** Diurnal mean wind speeds and directions between 2015 and 2018: (a) spring,

29 (b) summer, (c) autumn, and (d) winter. Note that the wind direction means the direction

30 the wind blows *from*.

31    **Table S1** | Seasonal-scale correlations (Pearson's *r*) between net ecosystem exchange ($r_{NEE}$), gross primary productivity ($r_{GPP}$), and ecosystem

32    respiration ($r_{Rec}$) and the key environmental factors. N represents the sample size. Abbreviations: SWC, soil water content; $T_{soil}$, soil temperature.

33    **, $p < 0.01$; *, $p < 0.05$.

34

| Environment factors | Spring | | | | Summer | | | | Autumn | | | | Winter | | | 2014-2018 | | | |
|---|---|---|---|---|---|---|---|---|---|---|---|---|---|---|---|---|---|---|---|
| | $r_{NEE}$ | $r_{GPP}$ | $r_{Rec}$ | N | $r_{NEE}$ | $r_{GPP}$ | $r_{Rec}$ | N | $r_{NEE}$ | $r_{GPP}$ | $r_{Rec}$ | N | $r_{NEE}$ | $r_{Rec}$ | N | $r_{NEE}$ | $r_{GPP}$ | $r_{Rec}$ | N |
| SWC at 10 cm | -0.167** | 0.158** | 0.138** | 314 | -0.063 | 0.167** | 0.225** | 369 | 0.185** | 0.330** | 0.442** | 418 | -0.004 | -0.004 | 379 | -0.254** | 0.370** | 0.380** | 1476 |
| SWC at 20 cm | -0.356** | 0.467** | 0.379** | 314 | -0.060 | 0.164** | 0.223** | 369 | 0.138** | 0.453** | 0.509** | 418 | -0.129 | -0.129 | 379 | -0.391** | 0.532** | 0.519** | 1476 |
| SWC at 30 cm | -0.391** | 0.565** | 0.533** | 314 | -0.037 | 0.170** | 0.273** | 369 | 0.163** | 0.465** | 0.541** | 418 | -0.255** | -0.255** | 379 | -0.433** | 0.609** | 0.610** | 1476 |
| SWC at 40 cm | -0.371 ** | 0.631** | 0.721** | 314 | -0.062 | 0.199** | 0.288** | 369 | 0.094 | 0.539** | 0.546** | 418 | -0.210** | -0.210** | 379 | -0.464** | 0.660** | 0.669** | 1476 |
| SWC at 50 cm | -0.338** | 0.623** | 0.768** | 314 | -0.168 ** | 0.318** | 0.347** | 369 | 0.091 | 0.515** | 0.523** | 418 | -0.209 ** | -0.209 ** | 379 | -0.484** | 0.688** | 0.696** | 1476 |
| $T_{soil}$ at 10 cm | -0.507** | 0.755** | 0.742** | 314 | -0.337** | 0.485** | 0.395** | 369 | 0.211** | 0.677** | 0.765** | 418 | -0.254** | -0.254** | 379 | -0.549** | 0.758** | 0.749** | 1476 |
| $T_{soil}$ at 20 cm | -0.515** | 0.770** | 0.760** | 314 | -0.310** | 0.494** | 0.457** | 369 | 0.211** | 0.678** | 0.766** | 418 | -0.314** | -0.314** | 379 | -0.545** | 0.758** | 0.753** | 1476 |
| $T_{soil}$ at 30 cm | -0.519** | 0.773** | 0.761** | 314 | -0.276** | 0.490** | 0.506** | 369 | 0.211** | 0.680** | 0.767** | 418 | -0.362** | -0.362** | 379 | -0.541** | 0.756** | 0.754** | 1476 |
| $T_{soil}$ at 40 cm | -0.524** | 0.778** | 0.763** | 314 | -0.244** | 0.490** | 0.557** | 369 | 0.209** | 0.682** | 0.767** | 418 | -0.369** | -0.369** | 379 | -0.535** | 0.753** | 0.755** | 1476 |
| $T_{soil}$ at 50 cm | -0.526** | 0.781** | 0.765** | 314 | -0.222** | 0.488** | 0.591** | 369 | 0.207** | 0.682** | 0.765** | 418 | -0.250** | -0.250** | 379 | -0.527** | 0.748** | 0.756** | 1476 |

35

---

## Referee Report (RR1)

Summary

This manuscript presents 5 years of eddy covariance data to quantify the carbon dynamics of a semiarid grassland ecosystem in China's Horquin Sandy Land. The authors examine variation in NEE, GPP, and Rec at several scales of aggregation, and examine the response of these fluxes to environmental drivers. The revised version now includes key references to the relevant literature. I appreciate that the authors have added hypotheses and goals to guide the analysis. However, these hypotheses are not formulated in a way that is testable given the available data. Moreover, the authors continue to draw inferences that are not readily supported by the evidence. With substantial modification of the study motivation, hypotheses, and the analysis/interpretation of data, these issues could be resolved in a way that advances understanding of carbon-water relations in this ecosystem.

Major comments:

A major aim (Goal 2) of this paper is to "explore the effects of changes in precipitation amount and frequency on seasonal and annual NEE, GPP, and Rec." Throughout the paper, the authors conclude that seasonality in fluxes was related to precipitation event size and frequency (e.g. L 28-29). It is not clear which evidence the authors use to draw this conclusion. Figure 7 shows that spring total precipitation explains some of the variation in spring carbon fluxes, but also shows that the relationships are weak for other months, such as summer, when peak GPP occurs.

Goal 2 is followed by the hypothesis that "an effective precipitation threshold would exist at around 5 mm, which could alter soil moisture in deeper layer and affect carbon fluxes in the sandy grassland ecosystem." Unfortunately, this hypothesis is not tested. In Figure 2, the authors explore how precipitation size translates into dynamics in soil moisture at various depths, but no connection is made to carbon fluxes.

The motivation and rationale for this study mostly relies on the novelty of the dataset. This happens several times in the introduction alone (L 64-66; L91-96; L 127-129). The authors have data to make a significant contribution, but the motivation for the study is not strongly articulated in the introduction.

There are sections in the results where the presentation of data is unclear, which makes it difficult to understand and interpret the study findings. For example, there are inconsistencies in the sign convention of Rec (Figs. 3-5). Additionally, in some places it is unclear how the data were used to generate figures (Figure 6). Below I discuss specific aspects of this.

Specific comments

Abstract

In the abstract, the authors conclude the ecosystem was an annual carbon source, and then present two possible explanations for why: because of drought, or due to a history of land

degradation. What I find problematic is that the abstract does not build toward a conclusion that is supported by evidence. Instead, two possible explanations are given that are untestable with the data. There is some evidence to support the conclusion that "drought [as quantified by low annual precipitation] decreased carbon sequestration (Figure 6-7), but there is no evidence in the paper to examine how land degradation influences carbon fluxes.

L 26-27: The statement "Annual precipitation had the strongest effect on annual NEE" is vague. I suggest modifying this sentence or combining it with the next one. For example, "Grassland carbon sequestration increased with increasing precipitation, as indicated by the dependency of NEE on annual precipitation."

L 27-28: The authors write "In the spring, NEE increased with increasing Tsoil and increasing precipitation. Is this a typo? Figure 7 shows that NEE actually decreased with increasing precipitation.

L 28-29: Please provide evidence for this.

L32-33: This was written in line 23.

Introduction

I am glad to see hypotheses in the revised version. However, in its current form, the first hypothesis cannot be tested with the available data. The analysis does not allow the authors to test for the effect of past land degradation on carbon sink activity. One way to rephrase this hypothesis is "we hypothesized that due to the strong dependence of GPP on precipitation in this ecosystem, years with low precipitation will be associated with carbon source activity."

Recommend the introduction be restructured to build toward a knowledge gap or hypothesis that is testable with the available data. Perhaps the introduction could be modified to explain why this study is necessary. Do the authors expect that what has been documented in other water-limited regions will not apply in the Horquin Sandy Land? If so, why?

L 56-61: I appreciate that the authors cite existing literature on carbon-water relations in drylands. However, instead of reporting previous findings, the introduction should synthesize content, identify a clear knowledge gap, and present hypotheses or study goals to address that gap. It is not clear how reporting prior work builds toward your study motivation. For example, what is the purpose of writing "evapotranspiration was a better proxy for the water available for NEE" if the authors then chose to use precipitation instead of ET to examine the response of NEE to water?

L 62-64: This is a good reference to the existing literature. Because drylands show a variety of source/sink behavior, there is need to study the Horquin Sandy Land.

L 101-105: There is a large body of work on how precipitation pulses drive carbon and water fluxes in semiarid regions, see below. See especially Figure 1 in Huxman et al. (2004).

L 105-108: Suggest reading Chen et al. (2009), who examined thresholds in a semiarid steppe ecosystem in Inner Mongolia: "The distinct responses of ecosystem photosynthesis and respiration to increasing pulse sizes led to a threshold in rain pulse size between 10 and 25 mm, above which post wetting responses favored carbon sequestration" (Chen et al., 2009).

L 113: Key reference missing: Noy-Meir (1973).

L 121-124: This statement about summer rainfall wets shallow soil layers seems to conflict with the statement in L147 that rain events greater than 5 mm wet deep layers.

Results

In the revised version, it is good to see that in section 3.1 the authors focus on key variables that drive observed dynamics in carbon fluxes. I appreciate the addition of panel d in Figure 3 to show annual total NEE, GPP, and Rec, and that Figure 4 now includes multiyear means in each panel. These figures now provide evidence to support the conclusion that the ecosystem was a carbon source at the annual scale.

Figures S1-S5: what do the error bars represent?

Figure S3: Why is SHF presented? It is not referenced elsewhere in the paper. Suggest remaking this figure with Tsoil. Diurnal patterns in Tsoil may help explain diurnal patterns in Rec, assuming soil respiration is a major component of Rec in this system.

Table 1: There should be a column header for "number of events" above "Magnitude of precipitation event". Also, since in L 271 the authors refer to the annual number of events, Table 1 should have a row with the annual amount of rainfall events for each size class.

Figure 2: It is not clear how this figure is used to draw conclusions. It shows that the size of a rain event influences dynamics in soil moisture at various depths. No connection is made to carbon fluxes, which is the main point of this paper. Additionally, the caption for Fig. 2 should define what the dashed line indicates.

Figure 3: Inconsistencies remain in the sign convention of carbon fluxes in figures. In Figure 3a-e, Rec is often negative, but it expressed as a positive cumulative total in panel f. This occurs again in Figures 4 and 5 whereas Rec fluxes are "large" when positive. Additionally, I suggest adding a zero line in Figure 3 (as in Fig. 5) to help the reader see the sign of carbon fluxes. There appear to be periods of negative GPP (e.g. 2015 DOY 250). Is this an error? Please explain.

Figures 4 and 5: what do the error bars represent?

Figure 6: It is unclear which data were used to make this figure. The caption says this is the relationship between annual precipitation and carbon fluxes. There are too many data points for these to be annual values. Also in L 346, if these are annual fluxes, why is precipitation maxing out at 100 mm, when annual precipitation ranged from 212-351 mm (L270)?

Figure 7: Typo for Autumn.

L 280: "Ecological links" is unclear; please explain.

L 283: Please provide a number for annual precipitation during a "normal year" to give context for how dry the experiment period was.

L325. I disagree that the diurnal pattern of Rec in summer was similar to that during spring. The diurnal patterns are essentially opposite. For b, why does peak Rec occur at night, instead of during the day when temperatures are highest? Is this related to the heating issue described in L436? Similarly, in Fig. 5d there are two peaks in NEE, and a minimum during the day. Does this pattern indicate some level of vegetation activity?

Discussion

L411: Specify as before the summer growing season.

L419-420: Please add a reference to figures to provide evidence in support of this statement. For example, Figure 8 shows that that these conditions increased carbon uptake (more negative NEE) because the sensitivity of GPP to Tsoil and moisture was greater than that of Rec (similar to the text in L 459-461).

L 429-430: "NEE decreased with increasing light intensity during the day." Are you referring to similar diurnal patterns of Rnet and NEE, or a light response of NEE? It is not surprising that NEE tracks the pattern of solar energy. I do not see why this result was included in the discussion.

L 435: Did the authors attempt to correct for these heating effects? How has this potential for sensor error influenced the interpretation of results? I think this warrants more discussion, given the study's emphasis on source/sink activity.

L 479-483: Instead of offering event size and frequency as a possible explanation for dynamics in NEE, what if you used data to test this idea? This would provide a direct test of the goal and hypothesis stated in the introduction. One way to test this is to calculate the mean or integrated fluxes corresponding to the times during which rain events of various sizes occurred. For example, in Table 1, rain events are grouped by size. Perhaps you could find a way to calculate the corresponding carbon dynamics for each of these rain groups. Such an exercise could inform results statements, such as "springs with a greater amount of 10-15 mm rain events had greater GPP than springs with fewer 10-15 mm rain events." Alternatively, you could order the seasons by integrated flux (e.g. total GPP) and rank years by number of large rain events, and test for a relationship between the two.

At a minimum, I think the authors should add precipitation to Figure 4. If a total precipitation bar was added to each of the years in the four subpanels, it would show if patterns in seasonal precipitation were related to variation in flux rates.

References

Huxman, T. E., Snyder, K. A., Tissue, D., Leffler, A. J., Ogle, K., Pockman, W. T., et al. (2004a). Precipitation pulses and carbon fluxes in semiarid and arid ecosystems, 254–268. https://doi.org/10.1007/s00442-004-1682-4

Noy-Meir, I. (1973). Desert Ecosystems: Environment and Producers. *Annual Review of Ecology and Systematics*, *4*(1), 25–51. https://doi.org/10.1146/annurev.es.04.110173.000325

Chen, S., Lin, G., Huang, J., & Jenerette, G. D. (2009). Dependence of carbon sequestration on the differential responses of ecosystem photosynthesis and respiration to rain pulses in a semiarid steppe. *Global Change Biology*, *15*(10), 2450–2461. https://doi.org/10.1111/j.1365-2486.2009.01879.x

Yan, L., Chen, S., Xia, J., & Luo, Y. (2014). Precipitation Regime Shift Enhanced the Rain Pulse Effect on Soil Respiration in a Semi-Arid Steppe. *PLoS ONE*, *9*(8), e104217. https://doi.org/10.1371/journal.pone.0104217

---

## Author Response (AR2)

26 October 2020

Dr. Trevor Keenan
Editor-in-Chief
*Biogeosciences*

RE: Submission of the revised manuscript (No. **bg-2020-89**): Variations in diurnal and seasonal net ecosystem carbon dioxide exchange in a semiarid sandy grassland ecosystem in China's Horqin Sandy Land.

Dear Dr. Trevor:

Thank you very much for your assistance in the review of our manuscript and for your invitation to resubmit our manuscript. We have revised the manuscript carefully according to reviewers' comments. We have also had this revised manuscript edited by Mr. Geoffrey Hart (ghart@videotron.ca/geoff@geoff-hart.com), an English science editor with more than 30 years of experience, to ensure that the quality of the language will be acceptable. Please contact him if necessary to confirm that he has performed this work or if you have any questions about the nature of the work that he has done. Our detailed responses to comments are presented in the remainder of this letter. All of revisions have been highlighted in red in the manuscript.

**Responses to Reviewers**

**Reviewer #1**
**Summary**
This manuscript presents 5 years of eddy covariance data to quantify the carbon dynamics of a semiarid grassland ecosystem in China's Horqin Sandy Land. The authors examine variation in NEE, GPP, and $R_{ec}$ at several scales of aggregation, and examine the response of these fluxes to environmental drivers. The revised version now includes key references to the relevant literature. I appreciate that the authors have added hypotheses and goals to guide the analysis. However, these hypotheses are not formulated in a way that is testable given the available data. Moreover, the authors continue to draw inferences that are not readily supported by the evidence. With substantial modification of the study motivation, hypotheses, and the analysis/interpretation of data, these issues could be resolved in a way that advances understanding of carbon-water relations in this ecosystem.

**Major comments:**
A major aim (Goal 2) of this paper is to "explore the effects of changes in precipitation amount and frequency on seasonal and annual NEE, GPP, and $R_{ec}$." Throughout the paper, the authors conclude that seasonality in fluxes was related to precipitation event size and frequency (e.g. L 28-29). It is not clear which evidence the authors use to draw this conclusion. Figure 7 shows that spring total precipitation explains some of the variation in spring carbon fluxes, but also shows that the relationships are weak for other months, such as summer, when peak GPP occurs.

We have revised Goal 2 as follows: "explore the effects of changes in total precipitation

and pulse size on NEE, GPP, and $R_{ec}$" (lines 143-144 in the revision). We have added Figures 2B, 2C, and 4 to provide data that let us test this goal. To further explore the relationship between summer precipitation and NEE and its components, we added the total seasonal precipitation to Figure 4 for each of the years in the four subpanels to show the relationships with precipitation. Figures 4 and 7 show that NEE, $R_{ec}$, and GPP were strongly controlled by the total summer precipitation (lines 32-35, 362-366 in the revision).

Goal 2 is followed by the hypothesis that "an effective precipitation threshold would exist at around 5 mm, which could alter soil moisture in deeper layer and affect carbon fluxes in the sandy grassland ecosystem." Unfortunately, this hypothesis is not tested. In Figure 2, the authors explore how precipitation size translates into dynamics in soil moisture at various depths, but no connection is made to carbon fluxes.
We have added Figures 2B and 2C to provide data that can be used to test the hypothesis (lines 370-385 in the revision).

The motivation and rationale for this study mostly relies on the novelty of the dataset. This happens several times in the introduction alone (L 64-66; L91-96; L 127-129). The authors have data to make a significant contribution, but the motivation for the study is not strongly articulated in the introduction.
We have deleted the duplicated descriptions of the novelty of the dataset, and have added our motivation for the study in the Introduction (lines 65-70, 89-93, and 125-127 in the revision).

There are sections in the results where the presentation of data is unclear, which makes it difficult to understand and interpret the study findings. For example, there are inconsistencies in the sign convention of $R_{ec}$ (Figs. 3-5). Additionally, in some places it is unclear how the data were used to generate figures (Figure 6). Below I discuss specific aspects of this.
We have revised the colors used in Figures 3-6 to be consistent with the sign convention for $R_{ec}$ and have confirmed that the NEE and GPP sign conventions are correct throughout the revision. We have clarified that Figure 6 shows the relationship between total monthly precipitation and total monthly NEE, $R_{ec}$, and GPP for the years with a complete dataset (2015, 2016, and 2018) (line 1074 and Figure 6 in the revision).

Specific comments
**Abstract**
In the abstract, the authors conclude the ecosystem was an annual carbon source, and then present two possible explanations for why: because of drought, or due to a history of land degradation. What I find problematic is that the abstract does not build toward a conclusion that is supported by evidence. Instead, two possible explanations are given that are untestable with the data. There is some evidence to support the conclusion that "drought [as quantified by low annual precipitation] decreased carbon sequestration (Figure 6-7), but there is no evidence in the paper to examine how land degradation influences carbon fluxes.

We have deleted the description of the effect of land degradation on carbon sink activity of grassland ecosystems. As you noted, our present data cannot support this conclusion.

L 26-27: The statement "Annual precipitation had the strongest effect on annual NEE" is vague. I suggest modifying this sentence or combining it with the next one. For example, "Grassland carbon sequestration increased with increasing precipitation, as indicated by the dependency of NEE on annual precipitation."
On the advice of our English editor, we added your suggested sentence, with some minor modifications (lines 28-30 in the revision).

L 27-28: The authors write "In the spring, NEE increased with increasing $T_{soil}$ and increasing precipitation. Is this a typo? Figure 7 shows that NEE actually decreased with increasing precipitation.
We had intended to say that the magnitude of NEE increased, so we have changed "increased" to "decreased" (line 31 in the revision).

L 28-29: Please provide evidence for this.
As we mentioned above, we have revised the major goal and have added evidence that can be used to test it (lines 32-35, 362-366, 379-384, Figures 2B, 2C, and 4).

L32-33: This was written in line 23.
We have deleted the duplicate description.

**Introduction**
I am glad to see hypotheses in the revised version. However, in its current form, the first hypothesis cannot be tested with the available data. The analysis does not allow the authors to test for the effect of past land degradation on carbon sink activity. One way to rephrase this hypothesis is "we hypothesized that due to the strong dependence of GPP on precipitation in this ecosystem, years with low precipitation will be associated with carbon source activity."
We have deleted the first hypothesis; as you note, it cannot be tested with the available data. We have then rephrased this hypothesis according to your comment, subject to some revisions by our English editor (lines 136-137 in the revision).

Recommend the introduction be restructured to build toward a knowledge gap or hypothesis that is testable with the available data. Perhaps the introduction could be modified to explain why this study is necessary. Do the authors expect that what has been documented in other water-limited regions will not apply in the Horquin Sandy Land? If so, why?
We have revised the Introduction to build toward the knowledge gap we designed the study to fill and describe a hypothesis that is testable with the available data. As noted earlier, we have deleted the description of the impact of land degradation on carbon fluxes, and focused on how the total precipitation amount and pulse size affected carbon fluxes in the Horqin Sandy Land. We have added a description of why this study is necessary (lines 65-70, 89-93, and 125-127 in the revision). Based on the existing research in other water-limited regions, we have proposed the hypothesis that the key

factors will be similar in the Horqin Sandy Land, and have verified this hypothesis using the data we collected (lines 136-137, 140-143, 144-147 in the revision).

L 56-61: I appreciate that the authors cite existing literature on carbon-water relations in drylands. However, instead of reporting previous findings, the introduction should synthesize content, identify a clear knowledge gap, and present hypotheses or study goals to address that gap. It is not clear how reporting prior work builds toward your study motivation. For example, what is the purpose of writing "evapotranspiration was a better proxy for the water available for NEE" if the authors then chose to use precipitation instead of ET to examine the response of NEE to water?
Citing previous studies in the Introduction is commonly done to provide context (what is already known, which leads to what is not known). However, we have removed our descriptions of previous findings in response to your comment, and have revised the description based on your comment (lines 59-70 in the revision).

L 62-64: This is a good reference to the existing literature. Because drylands show a variety of source/sink behavior, there is need to study the Horquin Sandy Land.
We have carefully read the paper and have cited the relevant results (lines 59-62 in the revision).

L 101-105: There is a large body of work on how precipitation pulses drive carbon and water fluxes in semiarid regions, see below. See especially Figure 1 in Huxman et al. (2004).
We have carefully read the paper and Figure 1, and have revised our description of how precipitation pulses drive carbon and water fluxes in semiarid regions (lines 98-100 in the revision).

L 105-108: Suggest reading Chen et al. (2009), who examined thresholds in a semiarid steppe ecosystem in Inner Mongolia: "The distinct responses of ecosystem photosynthesis and respiration to increasing pulse sizes led to a threshold in rain pulse size between 10 and 25 mm, above which post wetting responses favored carbon sequestration" (Chen et al., 2009).
We have carefully read the paper. Chen et al. (2009) studied the responses of photosynthesis and soil respiration under different water gradient treatments and compared the differences. However, our study was conducted under natural precipitation, similar to the study of Hao et al. (2010). Therefore, we have referred to the method of Hao et al. (2010) to support our belief that the effective precipitation threshold for changing the C flux in the sandy grassland ecosystem was 5 mm. Figure 2C supports this assumption.
Chen, S. P., Lin, G. H., Huang, J. H, Jenerette, G. D.: Dependence of carbon sequestration on the differential responses of ecosystem photosynthesis and respiration to rain pulses in a semiarid steppe. Glob. Change Biol., 15, 2450–2461. https://doi.org/10.1111/j.1365-2486.2009.01879.x, 2009.
Hao, Y. B., Wang, Y. F., Mei, X., and Cui, X. R.: The response of ecosystem $CO_2$ exchange to small precipitation pulses over a temperate steppe. Plant Ecol., 209, 335-347. https://doi.org/10.1007/s11258-010-9766-1, 2010.

L 113: Key reference missing: Noy-Meir (1973).
We have added the reference (lines 109, 865-866 in the revision).

L 121-124: This statement about summer rainfall wets shallow soil layers seems to conflict with the statement in L147 that rain events greater than 5 mm wet deep layers.
We have revised the description to clarify our meaning (lines 117-122 in the revision).

**Results**
In the revised version, it is good to see that in section 3.1 the authors focus on key variables that drive observed dynamics in carbon fluxes. I appreciate the addition of panel d in Figure 3 to show annual total NEE, GPP, and Rec, and that Figure 4 now includes multiyear means in each panel. These figures now provide evidence to support the conclusion that the ecosystem was a carbon source at the annual scale.
Thank you.

Figures S1-S5: what do the error bars represent?
We have added the meaning of the error bars (i.e., standard errors) for each supplemental figure (lines 15-16, 21-22, and 28-29).

Figure S3: Why is SHF presented? It is not referenced elsewhere in the paper. Suggest remaking this figure with $T_{soil}$. Diurnal patterns in $T_{soil}$ may help explain diurnal patterns in Rec, assuming soil respiration is a major component of $R_{ec}$ in this system.
We have revised this Figure to use $T_{soil}$, as you suggested (Figure S4 in the supplement).

Table 1: There should be a column header for "number of events" above "Magnitude of precipitation event". Also, since in L 271 the authors refer to the annual number of events, Table 1 should have a row with the annual amount of rainfall events for each size class.
We have deleted Table 1, because we changed one of our major goals to "explore the effects of changes in total precipitation and pulse size on NEE, GPP, and $R_{ec}$".

Figure 2: It is not clear how this figure is used to draw conclusions. It shows that the size of a rain event influences dynamics in soil moisture at various depths. No connection is made to carbon fluxes, which is the main point of this paper. Additionally, the caption for Fig. 2 should define what the dashed line indicates.
We have added Figures 2B and 2C to illustrate the response of carbon fluxes to precipitation pulses in different seasons (lines 370-385 in the revision), and also have defined the dashed line (lines 1045-1046 in the revision). Figure 2C tests the significance of differences in fluxes before and after an effective pulse to support our conclusion that these pulses were significant.

Figure 3: Inconsistencies remain in the sign convention of carbon fluxes in figures. In Figure 3a-e, $R_{ec}$ is often negative, but it expressed as a positive cumulative total in panel f. This occurs again in Figures 4 and 5 whereas $R_{ec}$ fluxes are "large" when positive. Additionally, I suggest adding a zero line in Figure 3 (as in Fig. 5) to help the reader

see the sign of carbon fluxes. There appear to be periods of negative GPP (e.g. 2015 DOY 250). Is this an error? Please explain.

As we mentioned above, we have revised the colors in Figures 3-6 to be consistent with the sign convention for $R_{ec}$ and have checked that the NEE and GPP sign conventions are correct throughout the revision. We have added a zero line in Figure 3. There were no negative GPP values in Figure 3, although the GPP value gradually decreases to a value near 0 at the end of the growing season. To display the data more clearly, we increased the size of the symbols, and that may have caused part of the symbol to extend below the zero line at GPP values near 0.

Figures 4 and 5: what do the error bars represent?

We have added the meaning of the error bars (i.e., standard errors) in the revision (lines 1062-1063, and 1073 in the revision).

Figure 6: It is unclear which data were used to make this figure. The caption says this is the relationship between annual precipitation and carbon fluxes. There are too many data points for these to be annual values. Also in L 346, if these are annual fluxes, why is precipitation maxing out at 100 mm, when annual precipitation ranged from 212-351 mm (L270)?

We have revised the caption as "Relationship between total monthly precipitation (PPT) and monthly net $CO_2$ flux" (lines 1074 in the revision), because we only had complete observations in 3 years, which was too little data to perform statistical tests. Instead, we have used the monthly data from 3 years to explore the influence of precipitation on carbon fluxes.

Figure 7: Typo for Autumn.

We have corrected the typo.

L 280: "Ecological links" is unclear; please explain.

We have revised "Ecological links" as "the ecosystem's carbon absorption and emission processes" (lines 277-278 in the revision).

L 283: Please provide a number for annual precipitation during a "normal year" to give context for how dry the experiment period was.

We have added the value for annual precipitation during a "normal year" (lines 281-282 in the revision).

L325. I disagree that the diurnal pattern of $R_{ec}$ in summer was similar to that during spring. The diurnal patterns are essentially opposite. For b, why does peak $R_{ec}$ occur at night, instead of during the day when temperatures are highest? Is this related to the heating issue described in L436? Similarly, in Fig. 5d there are two peaks in NEE, and a minimum during the day. Does this pattern indicate some level of vegetation activity?

We have revised the description according to your comment (lines 320-321, 328-329, and 335 in the revision), and have added an explanation for why peak $R_{ec}$ occurs at night in the summer (lines 458-467 in the revision). The reason for the two peaks in NEE (Fig. 5d) may be heating effects in the open-path infrared gas analyzer (lines 469-

487 in the revision).

**Discussion**

L411: Specify as before the summer growing season.

We have added "summer" in the growing season (line 436 in the revision).

L419-420: Please add a reference to figures to provide evidence in support of this statement. For example, Figure 8 shows that that these conditions increased carbon uptake (more negative NEE) because the sensitivity of GPP to $T_{soil}$ and moisture was greater than that of $R_{ec}$ (similar to the text in L 459-461).

We have cited the relevant figures to provide evidence in support of this statement (lines 447-449 in the revision).

L 429-430: "NEE decreased with increasing light intensity during the day." Are you referring to similar diurnal patterns of Rnet and NEE, or a light response of NEE? It is not surprising that NEE tracks the pattern of solar energy. I do not see why this result was included in the discussion.

What we wanted to express was that NEE responded to light. Based on your comment, we have moved this description to the Results (lines 325-328 in the revision).

L 435: Did the authors attempt to correct for these heating effects? How has this potential for sensor error influenced the interpretation of results? I think this warrants more discussion, given the study's emphasis on source/sink activity.

According to Goulden et al. (2006) and Burba et al. (2008), yearly estimates of NEE may be significantly biased toward $CO_2$ uptake in cold-climate ecosystems, so we attempted to correct for these heating effects (lines 216-217 in the revision). We have added a description of how the self-heating effect may have affected the results in the Discussion (lines 470-487 in the revision).

L 479-483: Instead of offering event size and frequency as a possible explanation for dynamics in NEE, what if you used data to test this idea? This would provide a direct test of the goal and hypothesis stated in the introduction. One way to test this is to calculate the mean or integrated fluxes corresponding to the times during which rain events of various sizes occurred. For example, in Table 1, rain events are grouped by size. Perhaps you could find a way to calculate the corresponding carbon dynamics for each of these rain groups. Such an exercise could inform results statements, such as "springs with a greater amount of 10-15 mm rain events had greater GPP than springs with fewer 10-15 mm rain events." Alternatively, you could order the seasons by integrated flux (e.g. total GPP) and rank years by number of large rain events, and test for a relationship between the two.

We tried to analyze the mean fluxes corresponding to the times during which rain events of various sizes occurred. However, because of the uneven distribution of precipitation in our semi-arid area, there were overlaps between events with different amounts of precipitation, so the effects of different levels of precipitation size on carbon flux could not be accurately determined based on the data we collected. Therefore, we have

modified the main goal to a goal that could be tested in the Introduction: to explore the effects of changes in total precipitation amount and pulse size on NEE, GPP, and $R_{ec}$. We have revised the description of how precipitation pulses affected the carbon fluxes and have added Figures 2B and 2C to provide the data (lines 32-35, 362-366, 379-384, Figure 2B and 2C, and Figure 4 in the revision).

At a minimum, I think the authors should add precipitation to Figure 4. If a total precipitation bar was added to each of the years in the four subpanels, it would show if patterns in seasonal precipitation were related to variation in flux rates.
We have added a bar for total precipitation for each of the years in the four subpanels of Figure 4 to show the relationship between seasonal precipitation and carbon fluxes (Figure 4, lines 362-366 in the revision).

References
Huxman, T. E., Snyder, K. A., Tissue, D., Leffler, A. J., Ogle, K., Pockman, W. T., et al. (2004a). Precipitation pulses and carbon fluxes in semiarid and arid ecosystems, 254–268. https://doi.org/10.1007/s00442-004-1682-4
Noy-Meir, I. (1973). Desert Ecosystems: Environment and Producers. Annual Review of Ecology and Systematics, 4(1), 25–51. https://doi.org/10.1146/annurev.es.04.110173.000325
Chen, S., Lin, G., Huang, J., & Jenerette, G. D. (2009). Dependence of carbon sequestration on the differential responses of ecosystem photosynthesis and respiration to rain pulses in a semiarid steppe. Global Change Biology, 15(10), 2450–2461. https://doi.org/10.1111/j.1365-2486.2009.01879.x
Yan, L., Chen, S., Xia, J., & Luo, Y. (2014). Precipitation Regime Shift Enhanced the Rain Pulse Effect on Soil Respiration in a Semi-Arid Steppe. PLoS ONE, 9(8), e104217. https://doi.org/10.1371/journal.pone.0104217.
Thank you for directing our attention to these references. We have carefully read these papers and have cited the most relevant ones (highlighted in red in the revision).

Thanks for your efforts to improve our manuscript. We hope that our replies and the resulting changes will be satisfactory, but we will be happy to work with you to resolve any remaining issues.

**Reviewer#3:**

General comments:

I reviewed this manuscript for the first time. I realized it has been improved a lot after taking the reviewers' comments in the first round. But I think some further improvements remain required before being publishable. See my advices.

1. L19-21,"Sandy grasslands are sensitive to climate change, yet the magnitudes, patterns, and environmental controls of their $CO_2$ flows are poorly understood", the expression is not backed up by the current literature. Generally, there are lots of studies of carbon fluxes over sandy grasslands worldwide, but for some specific regions, the expression may hold.

We have revised this to clarify that this is true for some specific regions, such as our study area (lines 23-24 in the revision).

2. L136, '$CO_2$ dynamics' -> '$CO_2$ fluxes'

We have revised this as "$CO_2$ fluxes" (line 129 in the revision).

3. L98, 'quantified the temporal variation', you actually quantified the $CO_2$ fluxes over different timescales.

We have revised "quantified the temporal variation" to "quantified the $CO_2$ fluxes over different timescales" (line 131 in the revision).

4. L183-187, the description of precision and accuracy are very confusing, μmol/m2/s is the unit of flux, but here the authors are evaluating the raw measurements of IRGA; also, the precision and accuracy should be on the raw measurement of 0.1 s timescale for a given 10 Hz EC. It is strange to discuss the topic for raw measurement but on the timescale (30min) for the averaged flux.

We apologize for not clearly describing the processing interval and measurement interval; we calculated 30-min means using a 10 Hz measurement interval, so we have revised the description to clarify this (lines 198-202 in the revision).

5. L165, the measurement sections were messed up. You could describe meteorological measurements soon after the experimental site, then describe EC measurement and flux calculation, quality control etc. That way, the method section may flow better.

Based on your comment, we have revised the order of the measurement sections (lines 172-191 in the revision).

6. L219, the EC system you used is an open path one, I am not aware of any requirement of lag correction. You need to detail it.

Time lags should be always compensated. The only exception is when an open-path analyzer is located very close to an anemometer or overlapping with it. However, this configuration is not recommended due to the important flow distortion effects that result from the presence of the analyzer. Note that the instruction manual for the EddyPro software supports this response (https://www.licor.com/env/support/EddyPro/home.html).

7. L225, avoid using not so necessary description: you do not need to remove data during power failure as no data can be stored as the data logger is also dead then, right? We have removed this description.

8. L226, it is more intuitive if you use umol/m2/s as the unit of carbon flux. We have revised the unit of carbon flux to be "umol $m^{-2}$ $s^{-1}$" (lines 222-223 in the revision).

9. L229-231, The definition of daytime or nighttime NEE seems not useful and breaks the flow of the paragraph. As I reviewed it progressively, this sentence can be moved to the following paragraph for data gap filling. Based on your comment, we have moved this sentence to the gap-filling paragraph (lines 228-230 in the revision).

10. L249-251, you may use a graph with Rn-G as x-axis and H+LE as y-axis to show the energy closure, which can be part of the supplementary information. We have added the graph you requested to show the energy closure as Supplementary Figure S1.

11. Fig.3-5, the y-axis can be named as 'CO$_2$ fluxes'. On the advice of our English editor, we have revised the axis to "CO$_2$ flux". (Fig. 3-5 in the revision).

12. L355, The discussion needs further improvement. For 4.1, a more accurate subsection title is required. This subsection is very long, but the information is very divergent. So is the second part of the discussion. The authors may re-write the discussion. The authors can consider what to discuss before writing, e.g., you can have a subsection with a title like 'comparison with other arid grassland ecosystem', in which you can discuss if your finding is different from others and what new knowledge you can bring. Also, this is a data driven research, a possible limitation of the study can give the readers some knowledge to how much degree the conclusion is subject to some uncertainty, e.g., data quality or data treatment. I suggest the authors articulate the discussion in a clearer way rather than lay them out like a twin of the result section. We have added subsection titles in the Discussion to divide the descriptions as you suggested (lines 403, 433-434, 492, and 518 in the revision). According to your comment and reviewer 2's comment, we have deleted the descriptions that are not supported by existing data in the Discussion and have added possible explanations for some main results (lines 447-449, 458-467, and 469-487 in the revision).

Thanks for your efforts to improve our manuscript. We hope that our replies and the resulting changes will be satisfactory, but we will be happy to work with you to resolve any remaining issues.

Sincerely,

Yuqiang Li, Ph.D.

Northwest Institute of Eco-Environment and Resources
Chinese Academy of Sciences
320 Donggang West Road, Lanzhou, 730000, China
Phone/Fax: 86-931-496-7219
E-mail: liyq@lzb.ac.cn